# And-1 coordinates with polymerase δ to regulate nucleotide excision repair and UVB-induced skin tumorigenesis

Shuyan Zhou[1,7], Yi Zhang[1,7] ✉, Zongzhu Li[1], Zhuqing Li[1], Patricia S. Latham [2], Yunxiao Meng[3,4], Wen Chen[2,5], Penghua Yang[1], Chunyan Hou[6], Junfeng Ma [6] & Wenge Zhu [1] ✉

The nucleotide excision repair (NER) pathway is the primary mechanism for removing UVB-induced photoproducts in mammals. While early steps of NER are well defined, the later step of gap-filling DNA synthesis remains incompletely understood. Here, we report And-1, a DNA replication and repair factor, as a critical regulator of this process. And-1 localizes to UV lesions, directly interacts with the catalytic subunit of DNA polymerase δ (p125), and promotes its recruitment to facilitate repair synthesis. In vitro, And-1 enhances p125 polymerase activity. Importantly, And-1 function in NER requires phosphorylation at T826, which strengthens its binding to both damaged DNA and p125. To evaluate its physiological relevance, we generated phosphorylation-deficient And-1 knock-in mice. These mice exhibited impaired NER and developed keratoacanthomas upon chronic UVB exposure. Collectively, our findings uncover And-1 as a pivotal factor in NER-mediated DNA repair and highlight its role in skin tumorigenesis.

Excessive unprotected exposure to solar ultraviolet radiation (UVR) is a well-established risk factor leading to skin tumorigenesis[1,2]. UVR consists of three wavelength ranges: UVA (320–400 nm), UVB (280–320 nm), and UVC (200–280 nm). Among these, UVA predominantly contributes to skin aging and chronic inflammation, driving photoaging and long-term tissue damage[3]. In contrast, UVB and UVC are more potent in directly inducing DNA lesions, such as cyclobutane pyrimidine dimers (CPDs) and pyrimidine (6-4) pyrimidone photoproducts (6-4PPs), which are mutagenic and can compromise genomic integrity if not effectively removed. Since natural UVC is entirely absorbed by the ozone layer[4], UVB becomes the primary driver of UV-induced photocarcinogenesis[5,6]. In mammals, the nucleotide excision repair (NER) pathway serves as the principal mechanism for

repairing UV-induced DNA damage, particularly CPDs and 6-4PPs, as well as bulky base adducts caused by chemical mutagens[7].

NER is a multi-step process that efficiently removes bulky DNA lesions[8]. The NER pathway begins with UV DNA lesion recognition, facilitated by protein complexes such as XPC-RAD23B in global genome NER or RNA polymerase in transcription-coupled NER. Following lesion recognition, the helicases XPB and XPD are recruited to the UV lesion site to unwind dsDNA, creating a repair bubble. The damaged strand is then excised through coordinated incisions at the 5' and 3' sides of the lesion by the endonucleases XPF–ERCC1 and XPG, resulting in the removal of a ~24–32 nucleotide DNA fragment. The resulting gap is subsequently filled in by DNA polymerases using the undamaged strand as a template and sealed by DNA ligase to restore genomic integrity[9].

[1]Department of Biochemistry and Molecular Medicine, George Washington University School of Medicine and Health Sciences, GW Cancer Center, Washington, DC, USA. [2]Department of Pathology, George Washington University School of Medicine and Health Sciences, Washington, DC, USA. [3]Laboratory & Molecular and Genomic Pathology, Department of Laboratory and Transfusion Services, The George Washington University Hospital, Washington, DC, USA. [4]Department of Pathology & Anatomical Sciences, University of Missouri Health Care (UMHC), Columbia, MO, USA. [5]Veterans Affairs Medical Center (VAMC), Washington, DC, USA. [6]Department of Oncology, Lombardi Comprehensive Cancer Center, Georgetown University Medical Center, Washington, DC, USA. [7]These authors contributed equally: Shuyan Zhou, Yi Zhang. ✉e-mail: scholarzy@163.com; wz6812@gwu.edu

As the late step in NER repair, gap-filling DNA synthesis by DNA polymerases is crucial for maintaining genomic stability[10]. However, the process of gap-filling DNA synthesis is complicated and not yet fully understood. This process involves two distinct pathways that contribute to the completion of gap-filling DNA synthesis[11]. Polε is responsible for approximately 50% of the synthesis on the leading strand and is rapidly recruited to the damage sites following excision[12]. The remaining 50% of repair synthesis on the lagging strand relies on Polδ and Polκ. Once repair synthesis is complete, XPG contributes to processing the DNA ends, followed by ligation to seal the final nick[13]. Mammalian pol δ contains four subunits, p125, p66, p50, and p12[14]. p125 and p50 form the catalytic core of DNA polymerase δ, while p66 promotes the recruitment of this catalytic core to UV lesion sites. p12 stabilizes the complex and ensures efficient lagging-strand repair synthesis during NER[15]. p125 also coordinates with other proteins in the NER pathway, such as proliferating cell nuclear antigen (PCNA), which functions as a sliding clamp to enhance the efficiency and processivity of DNA synthesis[16]. Although these polymerase subunits are critical for gap-filling DNA synthesis, the detailed regulatory mechanism governing the activity of these polymerases at UV lesion sites for initiating gap-filling synthesis remains poorly characterized.

And-1/WDHD1/Ctf4 is an acidic nucleoplasmic DNA-binding protein and is highly conserved from fungi to vertebrates[17]. And-1 is characterized by three major domains: an N-terminal WD40 repeat domain, which typically acts as a protein interaction scaffold; a central SepB domain, which specifically mediates protein–protein interactions; and a C-terminal HMG domain, which facilitates chromatin engagement through DNA binding[18]. Previous studies have highlighted And-1's critical roles in DNA replication[19], cell cycle progression[20], sister chromatid cohesion[21], homologous recombination (HR) repair[22,23], and DNA interstrand crosslink (ICL) repair[24]. Yeast cells with Ctf4 deletion are hypersensitive to UV light[25], suggesting that And-1/Ctf4 may be involved in the NER pathway. However, whether and how And-1 regulates NER remains unknown, and even less is known about the physiologic impact of And-1 deficiency in NER and tumorigenesis.

In this study, we identified a critical role of And-1 in the NER pathway. Specifically, And-1 accumulates at UV lesion sites in cells upon exposure to UVB. Inhibition of And-1 in human cells compromises the repair of photoproducts in DNA and reduces cell viability following UVB exposure. Moreover, And-1 facilitates gap-filling DNA synthesis by promoting the recruitment of Pol δ-p125 to UV lesion sites and enhancing its polymerase activity during gap-filling DNA synthesis. The role of And-1 in NER is dependent on its phosphorylation at T826, which creates a favorable conformation change that enhances its interaction with both UV lesion DNA and p125. Most significantly, we generated the And-1 phosphorylation-deficient mice (T819A, corresponding to T826 in human) and found these mice exhibit NER defects and increased susceptibility to keratoacanthomas under chronic UVB exposure, demonstrating a critical role of And-1 in NER and skin tumorigenesis.

## Results

### And-1 is a critical factor involved in repairing UVB-induced DNA lesions

To investigate whether or not And-1 is directly recruited to UV-induced DNA lesion sites, we first performed immunofluorescence (IF) analysis to examine the localization of And-1 in primary human epidermal keratinocytes (HEKa), which are covered with an isoprene polycarbonate membrane filter, and irradiated by UVB to induce the localized UV lesion in nuclei via pores of the membrane filter. Significantly, UVB treatment induced the accumulation of CPDs and 6-4PPs loci in nuclei, where And-1 was found to co-localize with both CPDs and 6-4PPs in HEKa cells (Fig. 1A–C). Consistently, we also observed the co-localization of And-1 with CPDs and 6-4PPs in response to UVB in both immortalized human keratinocyte cells

(HaCaT cells) and primary human fibroblast cells (Hs27 cells) (Supplementary Fig. 1A, B).

Next, we investigated whether And-1 is involved in repairing UV-induced DNA lesions. To this end, we performed ELISA to assess the removal efficiency of UVB-induced DNA lesions (CPDs and 6-4PPs) over the time following UVB treatment in HEKa, HaCaT and Hs27 cells. The results showed that And-1 depletion significantly compromised the capability of cells to remove both CPDs and 6-4PPs compared to siGL2 control-treated cells (Fig. 1D–J). Although the repair kinetics of CPDs and 6-4PPs exhibited slight variations among the different cell lines, CPD removal was consistently slower than that of 6-4PPs, which is in agreement with previous studies[6,26,27]. Moreover, And-1 depletion increased the sensitivity of HaCaT cells to UVB treatment, whereas ectopic expression of And-1 effectively restored cell viability (Fig. 1K and Supplementary Fig. 1C), ruling out the possibility of siRNA off-target effect.

These findings collectively demonstrate that And-1 is a critical factor involved in repairing UVB-induced DNA lesions and is physically accumulated at UV lesion sites in cells.

### And-1 interacts with NER factors and plays a critical role in recruiting Pol δ-p125 to UV lesion sites to facilitate repair synthesis

Given that NER is the primary pathway responsible for repairing UV-induced DNA damage, we predicted that And-1 may be involved in NER. To accomplish this hypothesis, we first performed mass spectrometry (MS) analysis to identify And-1-associated proteins involved in the NER. To exclude the possibility of non-specific chromatin-mediated interactions, we included ethidium bromide and DNase I in the lysis buffer to disrupt protein-DNA interactions. The MS analysis of FLAG-And-1 immunoprecipitants (IPs) identified several NER factors, including DDB1 and POLD1/p125 (Fig. 2A). Co-immunoprecipitation (Co-IP) assays confirmed that both endogenous and exogenous And-1 interacted with DDB1 and p125, and these interactions were significantly enhanced following UVB exposure (Fig. 2B, C).

DDB1 is a key factor to recognize UV lesions at the early stages of NER[28], while p125 is essential for the gap-filling DNA synthesis post UV lesion excision[11]. To determine which stages And-1 regulates NER, we first analyzed the dynamics of recruitment of DDB1, And-1, and p125 to chromatin following UVB exposure. Intriguingly, DDB1 rapidly accumulated at chromatin as early as ~5 min and diminished at ~1 h following UVB exposure, indicating that DDB1 functions at the early stage of NER. In contrast, And-1 and p125 began to accumulate onto chromatin at ~5–10 min, with the maximum accumulation of And-1 at 4 h and p125 at 8 h after UVB exposure (Fig. 2D), suggesting that And-1 may be involved in NER late stage. We therefore hypothesized that And-1 regulates gap-filling DNA synthesis by facilitating the recruitment of p125 rather than DDB1 to UV lesion sites. To test this hypothesis, we analyzed the chromatin association of DDB1 and p125 in cells subjected to And-1 depletion in response to UVB irradiation. The results showed that And-1 depletion significantly impaired the chromatin accumulation of p125 but not DDB1 in response to UVB irradiation, while ectopic expression of And-1 restored the chromatin association of p125 (Fig. 2E). Consistently, IF assays indicated that loss of And-1 markedly compromised the co-localization of p125 but not DDB1 with CPDs, while ectopic expression of And-1 rescued the co-localization of p125 with CPDs (Fig. 2F and Supplementary Fig. 2A–C). Additionally, the depletion of p125 did not affect the recruitment of And-1 to chromatin (Supplementary Fig. 2D). Previous studies have shown that Polε is essential for NER by promoting high-fidelity repair synthesis on the leading strand[29]. Interestingly, And-1 depletion did not affect the association of Polε with chromatin (Supplementary Fig. 2E). Taken together, these results strongly suggest that And-1 specifically facilitates the recruitment of p125 to UV lesion sites.

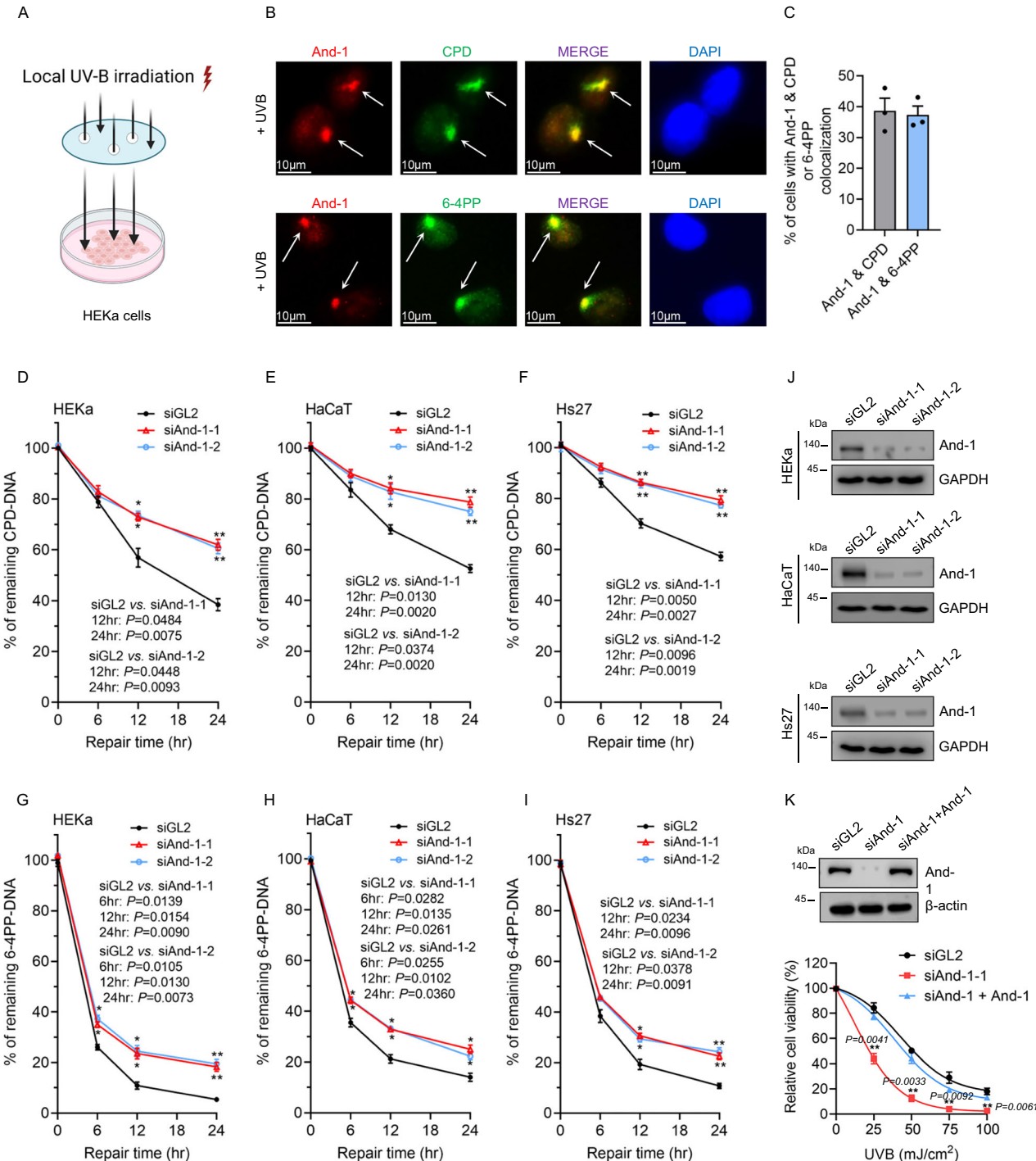

**Fig. 1 | And-1 is involved in NER in response to UVB irradiation. A** Schematic representation of UVB irradiation on HEKa cells shielded by an isoprene polycarbonate membrane filter with 5 μm diameter pores to generate localized UV lesion foci in nuclei. Created in BioRender. Zhou, S. (2025) https://BioRender.com/ssoy91s. **B** Immunofluorescence staining was used to examine the co-localization of And-1 with CPDs or 6-4PPs in response to UVB exposure in HEKa cells. **C** Quantification of the percentage of cells with co-localization of And-1 with CPDs or 6-4PPs. Approximately 100 cells from three independent fields were counted, and data are presented by mean ± SD. **D–I** HEKa (**D**, **G**), HaCaT (**E**, **H**) and Hs27 cells (**F**, **I**) were transfected with indicated siRNA for 40 h, followed by exposure to UVB at 40 mJ/cm². Cells were then harvested at the indicated time points, and genomic DNA was extracted and subjected to ELISA to examine the remaining levels of CPD-DNA and 6-4PP-DNA. Data are presented as mean ± SEM from three independent experiments (n = 3). **J** Cells shown in (**D–I**) were collected before UVB treatment, and cell lysates were analyzed by immunoblotting for the indicated proteins. **K** HaCaT cells were transfected with indicated siRNA and plasmid. At 24 h post-transfections, cells were exposed to UVB at indicated dosages. Cell viability was then measured 48 h after UVB irradiation, and data are presented as mean ± SEM from three independent experiments (n = 3). For (**D–I**) and (**K**), statistical analysis was performed by GraphPad Prism 9.0 using multiple two-tailed t-tests followed by Holm−Sidak correction. *P ≤ 0.05, **P ≤ 0.01. Source data are provided as a Source Data file.

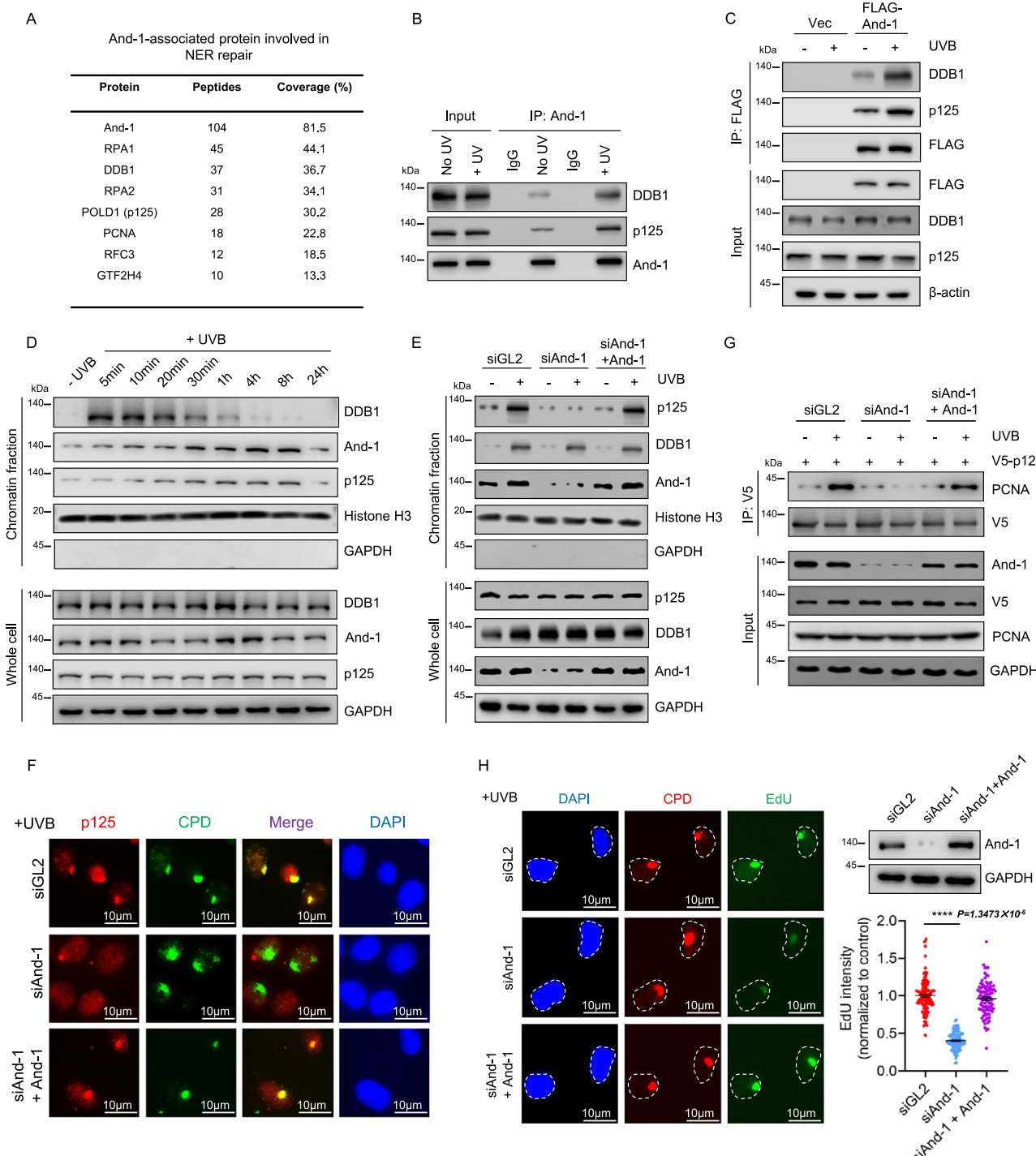

**Fig. 2 | And-1 plays a critical role in recruiting Polδ-p125 to UV lesion sites to facilitate repair synthesis. A** Mass spectrometry analysis to identify And-1-associated proteins involved in the NER. **B, C** Co-immunoprecipitation (co-IP) to detect the interaction of endogenous And-1 (**B**) or exogenous FLAG-And-1 (**C**) with indicated proteins in HEK293T cells (*n* = 3 independent experiments). **D** Chromatin fractions and whole-cell proteins were extracted from HaCaT cells at indicated time points following UVB exposure at 40 mJ/cm² and cell lysates were analyzed by immunoblotting for the indicated proteins (*n* = 3 independent experiments). **E** HaCaT cells were transfected with indicated siRNA and plasmid for 40 h, and cells were then exposed to UVB at 75 mJ/cm² and harvested 1 h post UVB irradiation. Chromatin fractions and whole-cell proteins were extracted and immunoblotted for indicated proteins (*n* = 3 independent experiments). **F** Immunofluorescence

staining was performed to examine the co-localization of p125 and CPDs in HaCaT cells exposed to UVB at 75 mJ/cm². **G** HEK293T cells were transfected with indicated siRNA and plasmid for 40 h, then harvested 1 h post UVB irradiation at 100 mJ/cm². V5-IP was performed in cell lysates, and then IPs were immunoblotted for indicated proteins. **H** HaCaT cells were transfected with the indicated siRNA and plasmid for 40 h, then exposed to UVB at 75 mJ/cm² and allowed to repair for 2 h. Unscheduled DNA synthesis (UDS) was then performed to assess Edu incorporation efficiency. DNA damage was identified by CPD staining. EdU levels were quantified by counting approximately 100 cells and normalized to control cells, *n* = 3 independent experiments. Data are presented as mean ± SEM. Statistical analysis was performed using unpaired two-tailed *t*-tests in GraphPad Prism 9.0. ***P ≤ 0.001. Source data are provided as a Source Data file.

We next investigated how And-1 regulates the recruitment of p125 onto UV lesion sites. Since PCNA is critical for the recruitment of p125 to UV lesion sites via protein–protein interactions[16], we therefore anticipated that And-1 may regulate p125 by affecting the interaction of p125 with PCNA. To test this possibility, we examined the interactions of p125 with PCNA in cells with And-1 depletion. Interestingly, UVB exposure increased the interaction of p125 with PCNA, and this interaction was significantly decreased in cells with And-1 depletion and ectopic expression of And-1 restored these interactions (Fig. 2G).

Since p66 is essential for the recruitment of p125 to UV lesion sites[11], we therefore examined whether And-1 interacts with p66 and regulates its localization to UV lesion sites. Co-IP result indicated that And-1 interacts with p66 (Supplementary Fig. 2F), and depletion of And-1 significantly reduced the co-localization of p66 with CPDs (Supplementary Fig. 2B, G, H), as well as the association of p66 but not PCNA with chromatin (Supplementary Fig. 2I). In line with these findings, we also observed that suppression of And-1 significantly impaired the chromatin association of p125 and p66, as well as their co-localization with CPDs in both HEKa and Hs27 cells, respectively (Supplementary Fig. 2J–Q).

To further validate the And-1's role in Pol δ–p125–mediated repair synthesis, we measured the unscheduled DNA synthesis (UDS) using the approach as previously described[30,31]. Specifically, HaCaT cells were locally irradiated with UVB and subsequently pulse-labeled with the thymidine analog 5-ethynyl-deoxyuridine (EdU) to evaluate DNA repair capacity. Robust EdU incorporation was detected at sites of localized UV-induced CPD lesions, and this incorporation was markedly reduced upon And-1 depletion, whereas ectopic expression of And-1 effectively restored EdU incorporation levels (Fig. 2H), indicating that And-1 plays an essential role in promoting repair synthesis during NER.

Taken together, these results suggest that And-1 facilitates the recruitment of Pol δ–p125 to chromatin by enhancing PCNA-p125 interaction and the recruitment of p66 to UV-lesion sites in response to UV exposure, thus promoting repair synthesis.

## Phosphorylation of And-1 at T826 is essential for its role in NER

Our previous studies revealed that phosphorylation of And-1 at T826 by ATR is critical for its role in DNA damage response[24,32]. To determine if this phosphorylation is also implicated in its role in NER, we examined the phosphorylation level of And-1 (p-And-1) upon UVB exposure. The results showed that p-And-1 levels were significantly elevated after UVB exposure, and this phosphorylation was effectively abolished upon treatment by VE-821, a known ATR inhibitor (ATRi)[33] (Supplementary Fig. 3A). Consistently, IF analysis demonstrated that p-And-1 co-localized with CPDs and VE-821 treatment significantly reduced the accumulation of p-And-1 at CPD loci (Fig. 3A and Supplementary Fig. 3B). Moreover, in And-1-depleted cells, ectopically expressed WT And-1 but not T826A And-1 was found to co-localize with CPDs (Fig. 3B and Supplementary Fig. 3C, D). These results indicate that And-1 phosphorylation at T826 is critical for its accumulation at UV lesion sites.

We next explored whether And-1 phosphorylation facilitates the recruitment of p125 to UV lesion sites. As shown in Supplementary Fig. 3E–G, reconstitution with WT And-1 but not the T826A mutant restored the co-localization of p125 with CPDs in And-1-depleted cells. Consistent with these findings, ectopic expression of WT And-1 but not T826A mutant significantly enhanced CPDs removal efficiency and promoted p125 chromatin association in And-1-depleted cells (Fig. 3C, D). Notably, the interaction of p125 with T826A mutant was significantly reduced compared to WT And-1 (Supplementary Fig. 3H). Reconstitution with WT And-1 but not the T826A mutant substantially restored the interactions of p125 with PCNA (Fig. 3E), indicating that phosphorylation of And-1 at T826 is critical to facilitate p125-PCNA interaction in cells exposed to UVB.

To further investigate whether phosphorylation of And-1 is required in repair synthesis, we performed UDS assays and found that

And-1 depletion markedly reduced EdU incorporation at CPD lesions, whereas ectopic expression of WT And-1 but not the T826A mutant restored this incorporation, indicating phosphorylation of And-1 at T826 is crucial for repair synthesis during NER (Fig. 3F–H).

We next sought to determine the role of ATR in regulating And-1's function during NER. To this end, we first assessed CPD removal efficiency in wild-type (shScramble) HaCaT cells with or without ATR inhibition using VE-821. ATR inhibition markedly delayed CPD clearance following UVB irradiation, indicating that ATR activity is critical for efficient NER. Notably, in And-1–depleted cells, CPD removal efficiency in T826A-expressing cells treated with ATRi was not significantly different from that in shScramble cells treated with ATRi, indicating that the T826A mutant does not confer additional defects beyond ATR inhibition. Importantly, expression of the phospho-mimetic T826E mutant in And-1–depleted cells partially but significantly restored CPD removal even in the presence of ATR inhibition, demonstrating that phosphorylation of And-1 at T826 is a primary target of ATR during NER (Fig. 3I).

To further validate these findings, we performed unscheduled DNA synthesis (UDS) assays using EdU incorporation under the same experimental conditions. Consistent with the CPD removal results, ATR inhibition significantly reduced EdU incorporation. UDS levels in T826A-expressing cells treated with ATRi were comparable to those in shScramble cells treated with ATRi. In contrast, expression of the T826E mutant partially but significantly restored EdU incorporation despite ATR inhibition (Supplementary Fig. 3I–K). Together, these results demonstrate that ATR-mediated phosphorylation of And-1 at T826 is essential for its function in NER.

## And-1 phosphorylation at T826 regulates its intramolecular interaction and association with UV lesion sites and p125

To understand how phosphorylation of And-1 regulates its role in NER, we generated a series of And-1 truncation constructs: full-length And-1 (FL, 1–1129), And-1 WD40 domain (1–330), And-1 SepB domain (330–984), And-1 HMG domain (984–1129), and And-1 SepB + HMG domain (330-1129). Co-IP assays revealed that p125 interacted with And-1 FL, And-1 (330–984), and And-1 (330–1129), indicating that the SepB domain (330–984) is critical for mediating the And-1-p125 interaction (Fig. 4A).

With these And-1 mutants, we next examined which And-1 domains are required for recruiting p125 to UV lesions using the IF assay. To this end, we expressed WT And-1 and And-1 mutants in cells with depletion of endogenous And-1 by shRNA, followed by examination of co-localization of p125 with CPDs. The results indicated that And-1 FL and And-1 (330–1129) but not other mutants successfully promoted the co-localization of p125 with CPDs in And-1-depleted cells (Fig. 4B). We further performed the UDS assay to determine the repair synthesis capacity of Polδ-p125 in the presence of these And-1 mutants. Consistently, only And-1 FL and And-1 (330–1129) mutants successfully restored Edu incorporation at the CPDs lesions in And-1-depleted cells (Supplementary Fig. 4A, B).

We next conducted chromatin fraction assays and found that And-1 FL, but not And-1 (1–984) mutant with depletion of HMG domain, could bind to chromatin (Fig. 4C). Consistently, And-1 FL but not And-1 (1–984) mutant promoted the chromatin association of p125 (Fig. 4C). These results suggest that the HMG domain is responsible for the association of And-1 with UV-lesion DNA, while the SepB domain mediates p125-And-1 interaction for its recruitment to UV lesion sites.

Previous studies have shown that And-1 phosphorylation at T826 prevents its intramolecular interaction between SepB and HMG domains in cells with crosslink DNA damage[24]. This observation led us to hypothesize that similar intramolecular conformation changes may occur in And-1 in response to UVB exposure. In line with this notion, we observed a sharp reduction in the interaction between the SepB and HMG domains following UVB exposure, coinciding with a significant

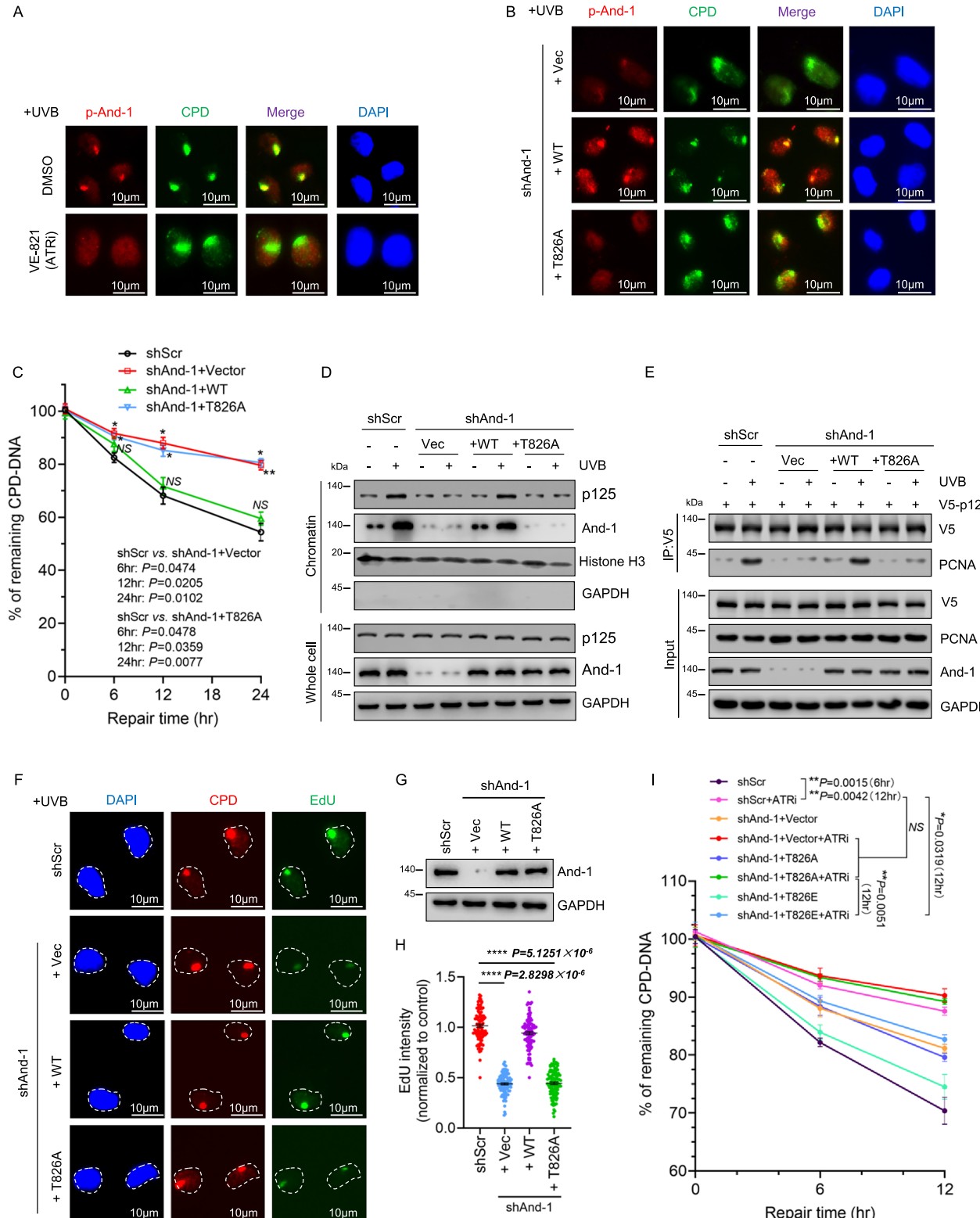

increase in p-And-1 levels (Fig. 4D). Moreover, treatment with increasing concentrations of ATRi (VE-821) gradually restored SepB-HMG binding and decreased p-And-1 level (Fig. 4D). Notably, UVB-induced disruption of the SepB-HMG interaction occurred only in the presence of the WT SepB domain, not the SepB domain with T826A mutation (Fig. 4E).

Collectively, the above results suggest that phosphorylation of And-1 at T826 disrupts the intramolecular interactions between SepB and HMG domains, allowing And-1 to bind UV lesion DNA via its HMG

domain and to interact with p125 through its SepB domain. This conformational change facilitates the recruitment of p125 to UVB lesion sites for gap-filling DNA synthesis (Fig. 4F).

### And-1 enhances the polymerase activity of p125 in vitro

During repairing UV lesions in NER, gap DNA and bubble structure are generated as intermediates following the unwinding of double-stranded DNA (dsDNA) by the TFIIH complex[34]. To further investigate the mechanism by which And-1 regulates NER in vitro, we

**Fig. 3 | Phosphorylation of And-1 at T826 is required for NER. A** HaCaT cells were pretreated with or without 5 μM ATRi (VE-821) for 6 h and then exposed to UVB irradiation at 75 mJ/cm². IF staining was performed to detect the co-localization of p-And-1 and CPDs (*n* = 3 independent experiments). **B** And-1-depleted HaCaT cells were transfected with the indicated plasmids for 40 h, then exposed to UVB, and IF staining was then performed to examine the co-localization of CPDs with p-And-1. **C** And-1-depleted HaCaT cells were transfected with indicated plasmids for 40 h, followed by exposure to UVB at 40 mJ/cm². Genomic DNA was then extracted at the indicated time points and subjected to ELISA to examine the remaining levels of CPD-DNA. Data are presented by mean ± SEM from three independent experiments. **D** And-1-depleted HaCaT cells were transfected with the indicated plasmids, then exposed to UVB at 75 mJ/cm². Cell lysates were then immunoblotted for the indicated proteins (*n* = 3 independent experiments). **E** And-1-depleted HEK293T cells were transfected with the indicated plasmids for 40 h, V5-IPs were harvested, and immunoblotted for the indicated proteins (*n* = 3 independent experiments). **F–H** And-1 depleted HaCaT cells were transfected with indicated plasmids for 40 h, then exposed to UVB at 75 mJ/cm². UDS assay was performed to assess Edu incorporation efficiency. Edu levels were quantified by counting approximately 100 cells and normalized to control cells. *n* = 3 independent experiments. Data are presented by mean ± SEM, and statistical analyses were performed by GraphPad Prism 9.0 using unpaired two-tailed *t*-tests for the data shown in (**H**). **I** And-1-depleted HaCaT cells were transfected with the indicated plasmids for 40 h, followed by pretreatment with or without 5 μM ATRi (VE-821) for 4 h. Cells were then exposed to UVB at 40 mJ/cm², and genomic DNA was extracted at the indicated time points and subjected to ELISA to examine the remaining levels of CPD-DNA. Data are presented by mean ± SEM from three independent experiments. For (**C**) and (**I**), statistical analyses were performed by GraphPad Prism 9.0 using the multiple two-tailed *t*-tests followed by Holm–Sidak correction. \**P* ≤ 0.05, \*\**P* ≤ 0.01, \*\*\**P* ≤ 0.001, "*NS*" indicates no significant difference. Source data are provided as a Source Data file.

generated biotin-labeled DNA structures, including single-stranded DNA (ssDNA), double-stranded DNA (dsDNA), gap DNA, and DNA bubble (Supplementary Fig. 5A). With these DNA molecules, we then performed DNA binding assays to determine whether And-1 directly binds to these structures by using in vitro co-IP assays. The results showed that purified recombinant And-1 exhibited a stronger affinity for ssDNA, DNA bubble, and gap DNA than dsDNA (Fig. 5A).

We next examined whether or not And-1 regulates the interactions of p125 with gap DNA in vitro. To this end, purified recombinant p125 proteins were incubated with And-1 together with gap DNA. Intriguingly, p125 alone displayed a weak affinity with gap DNA molecules, whereas the addition of And-1 proteins further enhanced the associations of p125 with the gap DNA (Fig. 5B and Supplementary Fig. 5B). Additionally, we mixed cell lysates from cells with or without And-1 depletion with gap DNA molecules and co-IP assay indicated that the association of p125 with gap DNA was significantly decreased upon And-1 depletion compared to control siRNA treatment, and p125 depletion by siRNA had no effect on the association of And-1 with gap DNA (Fig. 5C).

To assess whether And-1 phosphorylation at T826 modulates its interaction with the gap DNA and interaction of p125 with gap DNA in vitro, we purified recombinant WT And-1 or T826A mutant proteins from HEK293T cells with or without UVB irradiation. Interestingly, WT And-1 and T826A mutant purified from non-UVB-irradiated cells exhibited similar affinities for gap DNA. In contrast, WT And-1 but not the T826A mutant purified from UVB-exposed cells showed enhanced binding to gap DNA and significantly increased the association of p125 with gap DNA (Fig. 5D and Supplementary Fig. 5B). These results demonstrate that And-1 phosphorylation at T826 strengthens its interaction with gap DNA structures and facilitates the recruitment of p125 to gap DNA in vitro.

Since And-1 enhances the interaction of p125 with gap DNA in vitro, we then hypothesized that And-1 might promote its polymerase activity. To this end, we conducted an in vitro polymerase activity assay using a 125-nt template annealed with a 25-nt FAM-labeled DNA primer as described previously[35] (Fig. 5E). Purified p125 was incubated with increasing concentrations of WT And-1 or T826A mutant proteins, and DNA synthesis products were analyzed by denaturing PAGE. Interestingly, And-1 itself did not exhibit any polymerase activity, and p125 alone showed a baseline polymerase activity, as indicated by newly incorporating a maximum of ~25 nucleotides (top band ~50 nt), while most of the newly synthesized DNA products had ~5 nt nucleotides (bottom band, ~30 nt). The addition of WT And-1 with the increasing concentrations (10 nM, 25 nM, 50 nM) significantly increased the generation of newly synthesized DNA products with the sizes of 50 nt and 35 nt. Compared to the reaction with WT And-1, the DNA products of 50 nt and 35 nt in the reaction with the T826A mutant showed similar levels to those observed with p125 alone

(Fig. 5E). These results suggest that WT And-1, but not the T826A mutant, enhances both the catalytic activity and processivity of p125 in vitro.

## Phosphorylation of And-1 at T819 is critical for efficient NER in mouse skin epidermis

We next sought to determine whether And-1 phosphorylation plays a critical role in NER in vivo. To explore this possibility, we generated the *Wdhd1*^T819A (T819 is equivalent to T826 in humans) knock-in mice with C57BL/6J background (Supplementary Fig. 6A). Genotyping results confirmed the successful generation of *Wdhd1*^T819A knock-in mice by introducing an ACC-to-GCA mutation at the T819 site, and a silent mutation was also successfully introduced at the T816 site by changing TTG to CTT (Supplementary Fig. 6B). We examined the skin appearance of both *Wdhd1* WT and *Wdhd1*^T819A mice at 8-week and 34-week of age and observed no visible differences between these two groups, including the thickness of the epidermis (Supplementary Fig. 6C–E).

We then isolated epidermal keratinocytes from the dorsal skin of *Wdhd1* WT and *Wdhd1*^T819A mice at the age of 8 weeks and exposed these cells to UVB at 40 mJ/cm² as outlined in Fig. 6A. Chromatin fraction assays were then performed to evaluate whether And-1 phosphorylation at T819 influences the chromatin association of p125 following UVB exposure. As expected, in epidermal keratinocytes from *Wdhd1* WT but not *Wdhd1*^T819A mice, p-And-1 level was significantly increased following UVB treatment in both whole-cell and chromatin fractions, indicating And-1 phosphorylation was activated in epidermal cells by UVB exposure (Fig. 6B). Strikingly, the accumulation of p125 on chromatin was markedly elevated in WT epidermal cells but not in *Wdhd1*^T819A cells post UVB treatment (Fig. 6B), indicating that And-1 phosphorylation at T819 is critical for chromatin association of p125 in epidermal keratinocytes exposed to UVB. Furthermore, the CPDs and 6-4PP removal efficiency were significantly impaired in *Wdhd1*^T819A epidermal keratinocytes compared to WT epidermal cells (Fig. 6C).

To further assess whether And-1 phosphorylation of T819 is involved in the regulation of NER in mice, we exposed the dorsal skin of both *Wdhd1* WT and *Wdhd1*^T819A mice to a single dose of UVB (230 mJ/cm²) and collected skin samples at 1-, 8-, and 24-h post-irradiation for CPD-IHC. CPD levels were semi-quantitatively assessed by H-score, which combines staining intensity and percentage of CPD-positive cells. As shown in Fig. 6D, E, 1 h after irradiation, both *Wdhd1* WT and *Wdhd1*^T819A mice exhibited similar CPDs levels in their epidermis (*Wdhd1* WT: 222.84 ± 1.93, *Wdhd1*^T819A: 225.21 ± 1.59). However, at 8-h post-UVB exposure, the CPD levels in the epidermis of *Wdhd1* WT mice (148.07 ± 2.41) were markedly lower compared to those in *Wdhd1*^T819A mice (189.60 ± 2.20). The H-score for CPD at 24-h post-UVB exposure remained significantly higher in the *Wdhd1*^T819A group (126.69 ± 2.81) than that in the *Wdhd1* WT group (70.89 ± 1.47), suggesting impaired CPD removal in the epidermis of *Wdhd1*^T819A mice.

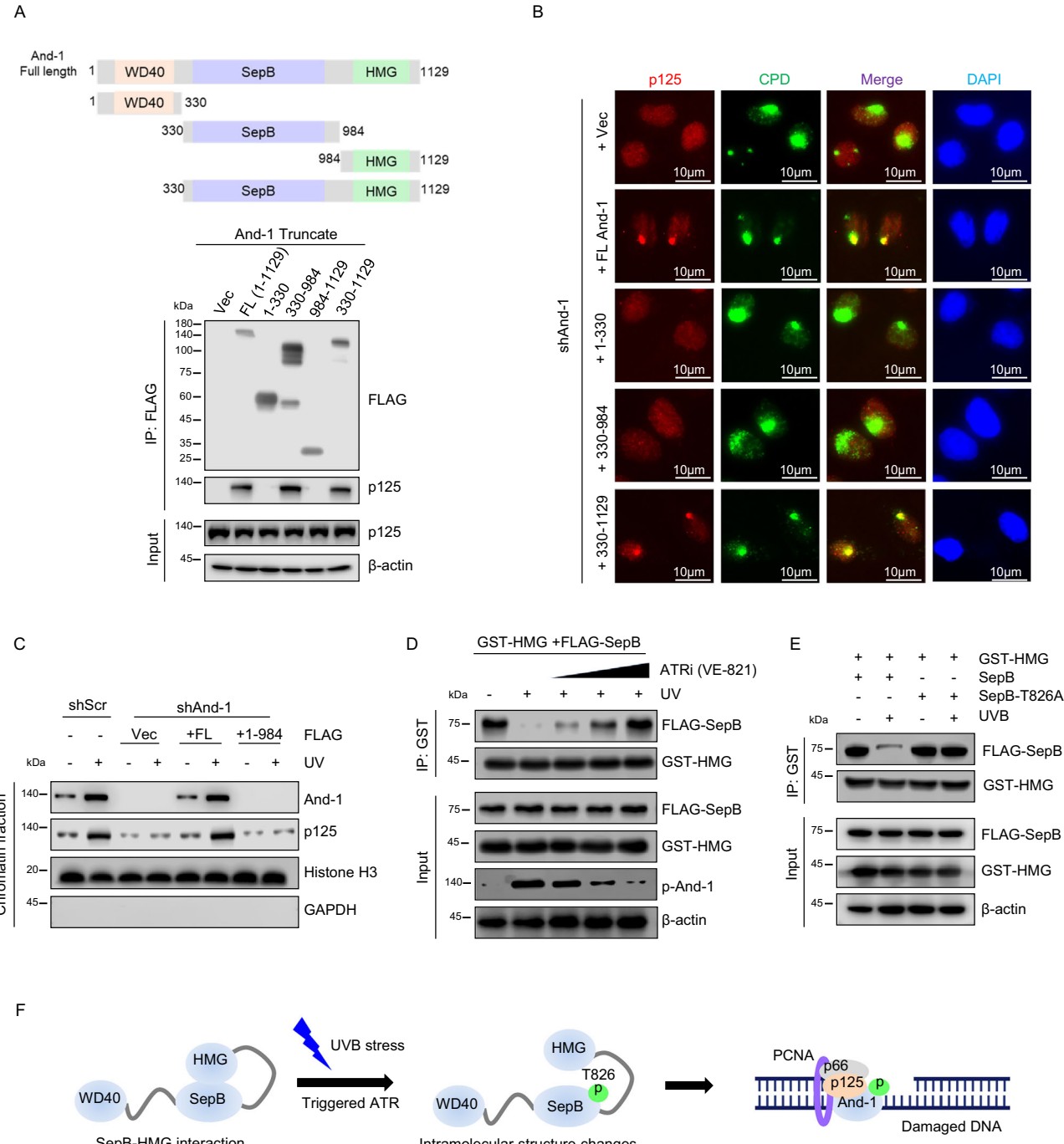

**Fig. 4 | And-1 phosphorylation at T826 regulates its intramolecular interaction and accumulation at UV lesion sites. A** Upper panel: Schematic representation of And-1 protein domains. Lower panel: FLAG-IPs of FLAG-And-1 and And-1 truncated mutants expressed in HEK293T cells, followed by immunoblotting for the indicated proteins ($n = 3$ independent experiments). **B** HaCaT cells with And-1 depletion were transfected with the plasmids expressing indicated And-1 or And-1 mutants. Cells were then harvested 1 h after exposure to UVB at 75 mJ/cm². Immunofluorescence staining was performed in harvested cells to examine the co-localization of p125 with CPDs ($n = 3$ independent experiments). **C** HEK293T cells with And-1 depletion by shRNA were transfected with plasmids expressing And-1 FL (1–1129) and And-1 (1–984) for 40 h. Cells were then harvested 1 h after exposure to UVB at 100 mJ/cm². Chromatin fractions were extracted and immunoblotted for indicated proteins ($n = 3$ independent experiments). **D** HEK293T cells were co-transfected with

FLAG-SepB and GST-HMG plasmids for 40 h. Then, cells were pretreated with or without ATR inhibitor (VE-821) for 6 h, with or without subsequent UVB irradiation at 100 mJ/cm². After 1-h repair period, cells were collected for further analysis. GST-IPs were performed in harvested cells, and IPs and inputs were then immunoblotted for the indicated proteins ($n = 3$ independent experiments). **E** HEK293T cells were co-transfected with the indicated plasmids for 40 h and then harvested 1 h after UVB exposure at 100 mJ/cm². GST-IPs were performed in harvested cells, and IPs and inputs were then immunoblotted for the indicated proteins ($n = 3$ independent experiments). **F** Schematic representation of And-1 phosphorylation prevents intramolecular interaction between SepB and HMG domains, thus facilitating the recruitment of p125 to gap DNA-UV lesion site by direct protein–protein interaction. Source data are provided as a Source Data file.

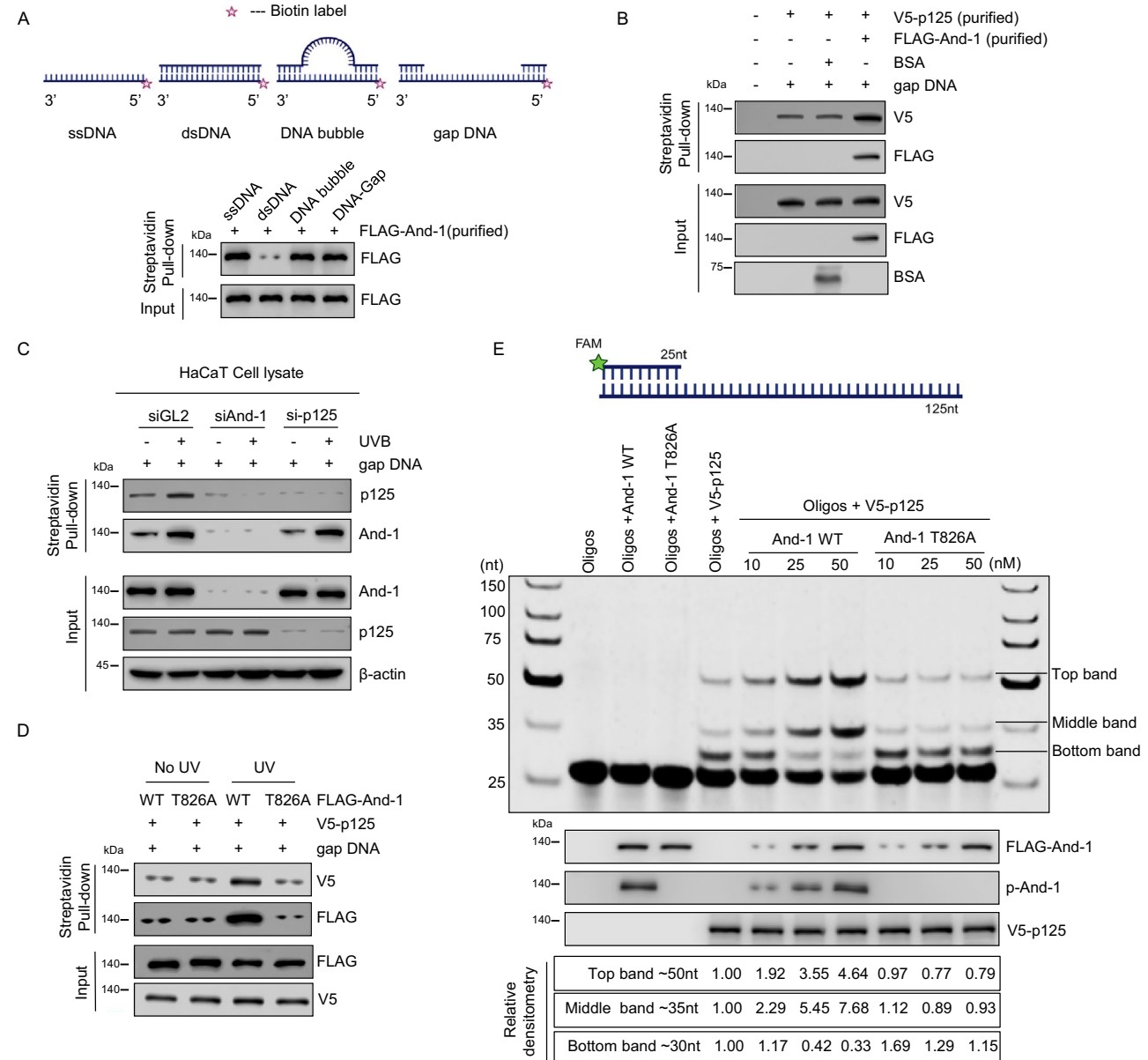

**Fig. 5 | And-1 promotes the association of p125 with gap DNA in vitro. A** Upper panel: Schematic representation of ssDNA, dsDNA, bubble DNA, and gap DNA, with biotin labeled at the 5' end of strand. Created in BioRender. Zhou, S. (2025) https://BioRender.com/509ij7n. Lower panel: immobilized 100-mer ssDNA, dsDNA, DNA bubble, and gap DNA were incubated with 400 ng of recombinant FLAG-And-1 for 45 min at room temperature. Streptavidin beads were added to the mixture and incubated at 4 °C overnight. The beads were washed, and bound proteins were resolved by SDS-PAGE, followed by immunoblotting for the indicated proteins (*n* = 3 independent experiments). **B** Immobilized 100-mer gap DNA was incubated with BSA (200 ng) or FLAG-And-1 (200 ng) for 45 min at room temperature, followed by incubation with recombinant V5-p125 (200 ng) at 4 °C overnight. The beads were then washed and processed as described in (**A**) (*n* = 3 independent experiments). **C** HaCaT cells were transfected with the indicated siRNA for 40 h and

then harvested 1 h after UVB irradiation at 75 mJ/cm². Cell lysates were incubated with the immobilized 100-mer gap DNA for 45 min at room temperature. The beads were then processed as described in (**A**) (*n* = 3 independent experiments). **D** HEK293T cells were transfected with WT And-1 or T826A mutant And-1 plasmids for 48 h and then treated with or without UVB at 100 mJ/cm². FLAG-And-1 (WT and T826A) were purified, and then 200 ng proteins were incubated with a 100-mer gap DNA for 45 min at room temperature, followed by incubation with recombinant V5-p125 (200 ng) at 4 °C overnight. The beads were then processed as described in (**A**) (*n* = 3 independent experiments). **E** Phosphorylated And-1 enhances p125 polymerase activity in a cell-free system (*n* = 3 independent experiments). The detailed experimental conditions are provided in the "Methods" section. Source data are provided as a Source Data file.

Collectively, these results indicate that And-1 phosphorylation at T819 is crucial for efficient NER in mouse skin epidermis.

## Mice with And-1 phosphorylation deficiency exhibit increased susceptibility to UVB-induced skin tumorigenesis

To investigate the role of And-1 phosphorylation at T819 in skin tumorigenesis, we subjected both *Wdhd1* WT and *Wdhd1*[T819A] mice to UVB irradiation on the dorsal skin, as depicted in Fig. 7A. Twenty-five

weeks after UVB exposure, all *Wdhd1*[T819A] mice developed multiple tumors with various sizes and numbers in UV-exposed areas, whereas only one of *Wdhd1* WT mice developed a single small-size tumor (Fig. 7B). Both the number and size of tumors were significantly increased in *Wdhd1*[T819A] mice compared to the WT mice group (Fig. 7C, D).

To characterize the skin tumors, we conducted H&E staining and IHC staining for Ki-67 on sections of mouse skin and tumor samples.

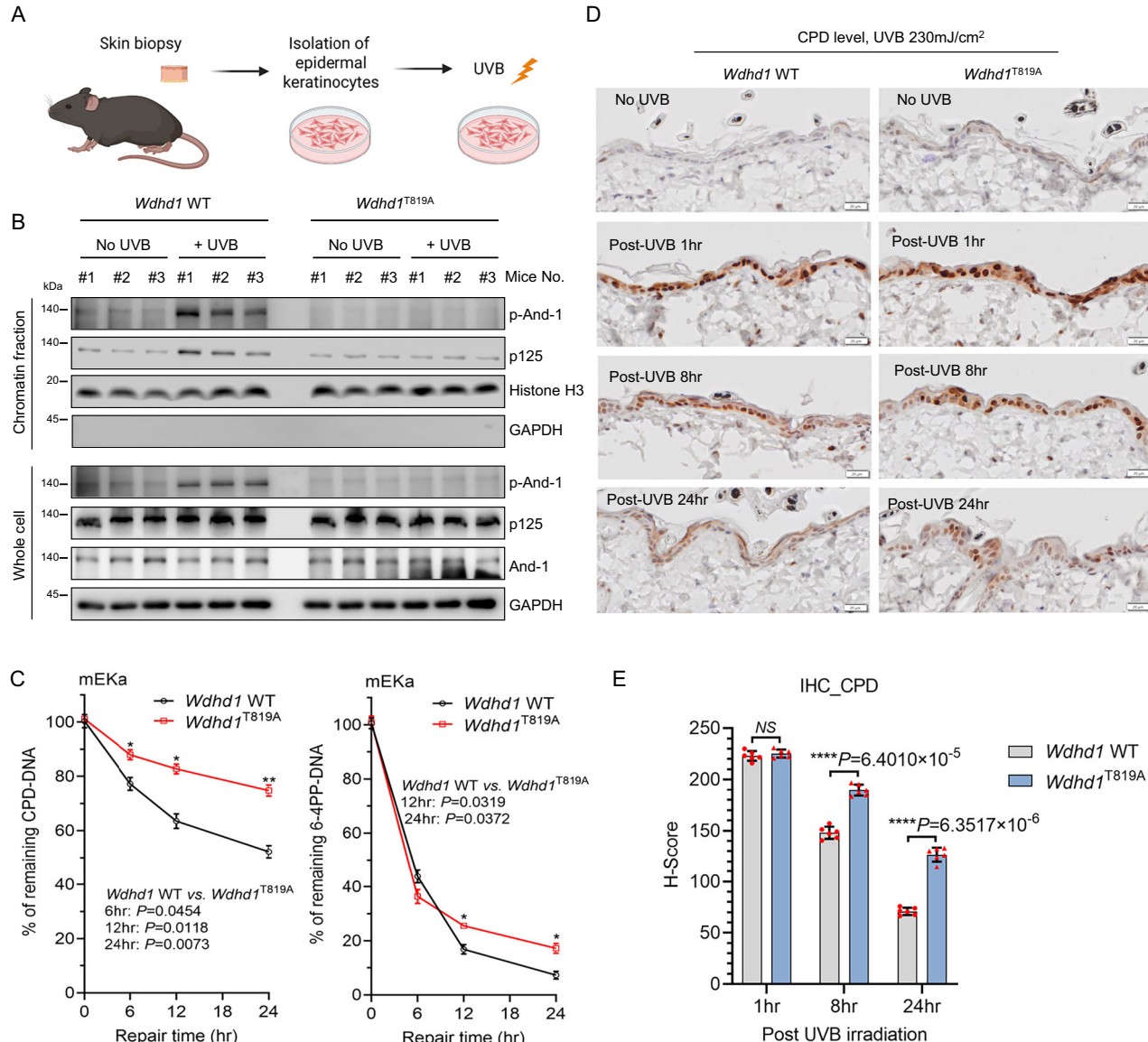

**Fig. 6 | And-1 phosphorylation is critical for efficient NER in mouse skin epidermis. A** The schematic representation of the isolation of epidermal cells from mouse skin biopsies. Created in BioRender. Zhou, S. (2025) https://BioRender.com/y5r6v2c. **B** Skin epidermal cells from *Wdhd1* WT and *Wdhd1*T819A mice were exposed to UVB radiation at a dose of 40 mJ/cm² and then harvested 1 h post-irradiation. Chromatin fractions and whole-cell lysates were extracted from harvested cells and then immunoblotted for the indicated proteins. **C** Skin epidermal cells from *Wdhd1* WT and *Wdhd1*T819A mice were exposed to UVB radiation at a dose of 40 mJ/cm² and harvested at indicated time points post-irradiation. Genomic DNA was extracted and subjected to ELISA to examine the remaining levels of CPD-DNA and 6-4PP-DNA. Data are presented as mean ± SEM from three independent experiments, and statistical analyses were performed using multiple two-tailed *t*-tests followed by

Holm−Sidak correction. **D** The dorsal skin of both *Wdhd1* WT and *Wdhd1*T819 mice was irradiated with UVB at a dose of 230 mJ/cm², and then the skin tissues were collected at the indicated time points for immunohistochemical (IHC) staining to assess CPD levels; scale bar is 20 μm. **E** H-score analysis was used to quantify the CPD levels as shown in (**D**), with a specific focus on the epidermal layer. The detailed calculation method was described in the "Methods" section. Data were presented as mean ± SEM. For each time point and group, *n* = 6 mice (independent cohorts) were analyzed, and statistical analyses were performed using unpaired two-tailed *t*-tests with Holm−Sidak correction. *\*P* ≤ 0.05. *\*\*P* ≤ 0.01, *\*\*\*\*P* ≤ 0.0001, "*NS*" indicates no significant difference. Source data are provided as a Source Data file.

Histological analysis revealed that all skin tumors from both the *Wdhd1*T819A group (mouse#1-5) and *Wdhd1* WT group (mouse#2) were diagnosed as keratoacanthomas (KAs)[36], with different stages of progression (Supplementary Data 3). The sole tumor from *Wdhd1* WT group mouse #2 is a KA at an early proliferating stage, characterized by a proliferating and thickened epidermal layer that is forming invaginations filled with excessive keratin while maintaining an intact and smooth basilar layer (Fig. 7Ea, Supplementary Fig. 7B1, B2). The other four *Wdhd1* WT mice skin samples, exemplified by mouse #5 (Fig. 7Eb), displayed only hyperkeratosis of the squamous epithelium,

accompanied by varying degrees of spinous layer thickening (Supplementary Fig. 7A1, A2, C1−E2). In contrast, all *Wdhd1*T819A mice developed relatively larger and more advanced KAs (Supplementary Fig. 7F1−K2), as exemplified by mouse #3 (Fig. 7Ec). This KA demonstrated a characteristic crater-like architecture with a prominent central keratin plug and showed thickening of the surrounding epidermis accompanied by parakeratosis. Moreover, the epidermal infolding of the lesion shows a proliferation of basilar cells and an increase in mitoses that remain in the basilar layer. Importantly, the area (indicated by the black arrow) at the base of the lesion exhibited an

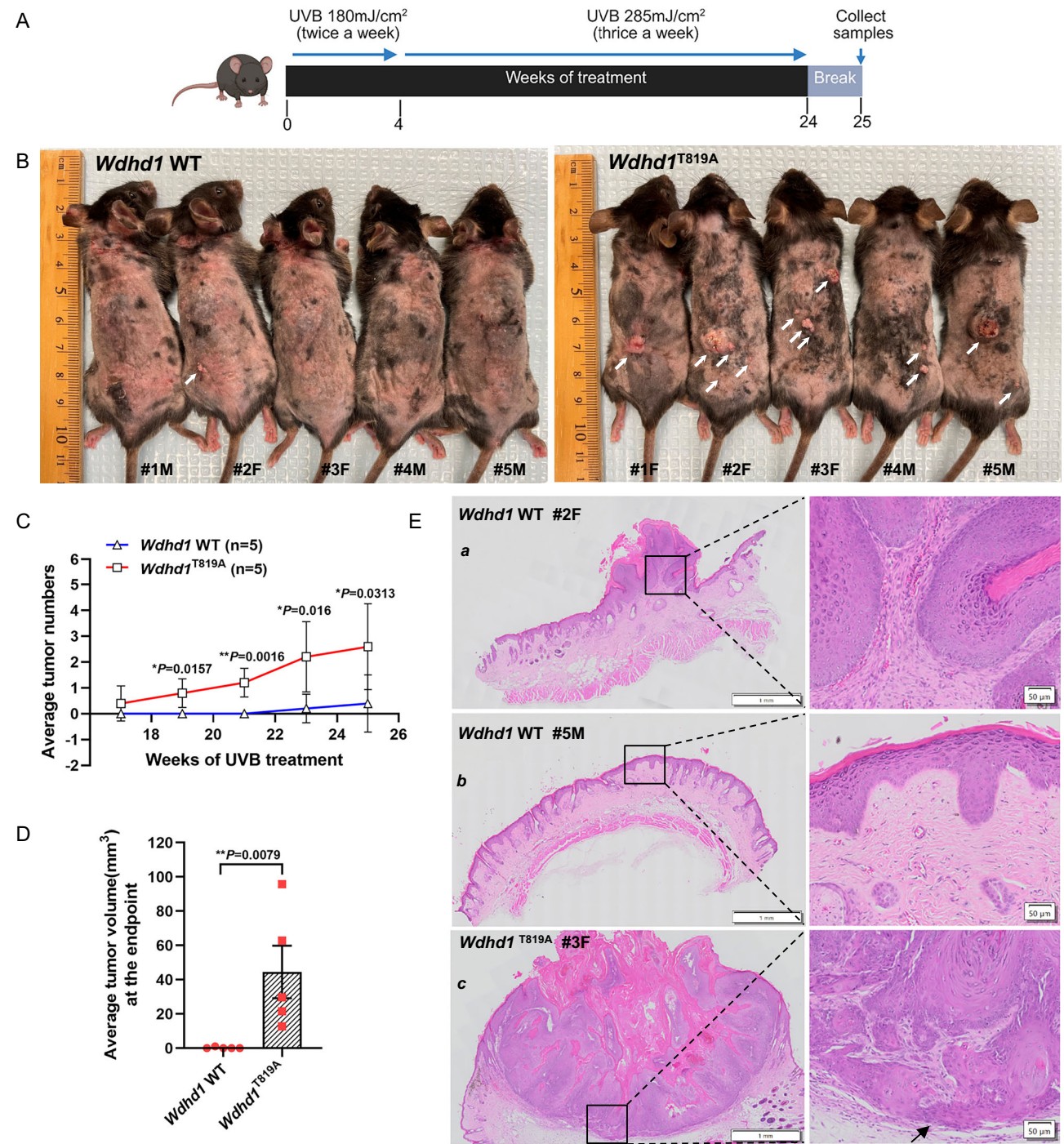

**Fig. 7 | And-1 phosphorylation-deficient mice exhibit increased susceptibility to UVB-induced skin tumorigenesis. A** Schematic representation of chronic UVB irradiation in mice to induce skin tumorigenesis. Created in BioRender. Zhou, S. (2025) https://BioRender.com/5kut4k3. **B** Representative images of skin tumors of both *Wdhd1* WT and *Wdhd1*^T819A^ mice induced by UVB irradiation at the treatment endpoint (week 25) (*n* = 5). **C** Number of tumors observed in both *Wdhd1* WT and *Wdhd1*^T819A^ mice under UVB treatment (*n* = 5). Data are represented as mean ± SEM, and statistical analyses were performed by GraphPad Prism 9.0 using a two-tailed Mann–Whitney U test, followed by Holm-Bonferroni correction. **D** Tumor volumes observed in both *Wdhd1* WT and *Wdhd1*^T819A^ mice at the endpoint (week 25) (*n* = 5). Data are represented as mean ± SEM, statistical analysis were performed by GraphPad Prism 9.0 using Welch's unpaired two-tailed *t*-test. **E** H&E staining of keratoacanthoma sections from *Wdhd1* WT (represented by mice #2 and #5) and *Wdhd1*^T819A^ mice (represented by mice #3), respectively. The scale bar in the left panel represents 1 mm. The right panel showed zoomed-in images of the square region indicated in the left panel; the scale bar is 50 μm. Black arrows indicated the micro-invasion areas of tumor. M male, F female. *P ≤ 0.05, **P ≤ 0.01. Source data are provided as a Source Data file.

infiltrative growth pattern, suggesting a potential malignant progression in this lesion (Fig. 7Ec and Supplementary Fig. 7H1, H2).

Given the increased pigmentation observed in the skin of *Wdhd1*^T819A^ mice (Fig. 7B), we performed IHC staining (AP/Fast Red detection) for SOX10, Melan-A, and Ki-67 on available *Wdhd1*^T819A^ tumor sections from the pigmented areas to evaluate the possibility of melanoma. The stained sections showed focal SOX10 positivity, rare Melan-A expression, and very low Ki-67 proliferation, with nuclei displaying normal morphology and no atypia (Supplementary Fig. 8A–E). These histopathological and immunohistochemical findings support a

diagnosis of benign post-inflammatory or UV-induced melanotic pigmentation, rather than melanoma.

Collectively, these in vivo results suggest that And-1 phosphorylation at T819 is crucial in the regulation of UVB-induced skin tumorigenesis.

## Discussion

In this study, we demonstrate that And-1 plays a pivotal role in NER by facilitating gap-filling DNA synthesis at UV lesion sites through coordinating with p125, p66, and PCNA. The role of And-1 in NER relies on its phosphorylation at T826. Both cell-based study and mouse models demonstrated that And-1 deficiency hinders the removal of UVB-induced photoproducts. This functional defect was further supported by reduced unscheduled DNA synthesis (UDS) upon And-1 depletion, indicating impaired NER-associated repair synthesis. Moreover, And-1 phosphorylation-deficient mice exhibit a significantly increased susceptibility to keratoacanthomas under chronic UVB exposure, emphasizing the critical role of And-1 in maintaining the integrity of the NER pathway to prevent UV-induced skin tumorigenesis.

Although the dual incision was proposed to act as an early step ahead of gap-filling synthesis in NER[37–39], recent evidence also suggests that the 5' incision by ERCC1-XPF precedes the initiation of repair synthesis, while the 3' incision by XPG may occur after or concurrently with repair synthesis, potentially acting to remove a DNA flap before final ligation[11,13]. In And-1-depleted cells, CPDs accumulate at UV lesion sites, where gap-filling synthesis is compromised (Fig. 2H), indicating that impaired gap-filling synthesis due to And-1 depletion may compromise 3' incision, which is in favor of the 5' and 3' sequential incision model. In this study, we did not measure the exact timing of 5' incision, 3' incision and gap-filling synthesis. Further detailed in vitro assays may be needed to test how And-1 bridges the early 5' incision and subsequent 3' incision via gap-filling synthesis.

Our data showed that And-1 interacts with p125 via its SepB domain (Fig. 4A), and And-1 inhibition disrupts the interactions between p125 and PCNA (Fig. 2G) but does not affect the chromatin loading of PCNA (Supplementary Fig. 2I). These findings suggest that And-1 appears to be involved in the assembly of protein complexes for gap-filling DNA synthesis. Since proteins with the HMG domain usually facilitate the formation of nucleoprotein complexes on the chromatin by bending DNA structure[40], it is not surprising to see that And-1 also participates in the assembly of the protein complex at UV lesion sites in NER. So far, we have not yet known whether and how And-1 changes DNA structure at UV lesion sites to facilitate the assembly of the DNA polymerase complex at UV lesion sites. Future studies using structural analysis are expected to elucidate how the HMG-domain protein And-1 affects DNA structure during gap-filling DNA synthesis in NER.

Phosphorylation of And-1 at T826 prevents the interaction between SepB and HMG domains following UVB exposure, thus facilitating its accumulation to UV lesion DNA via the HMG domain and enhancing its interaction with p125 through the SepB domain for the recruitment of p125 to lesion sites. This observation is in agreement with the previous study that And-1 T826 phosphorylation is crucial for its localization to damage sites and the subsequent recruiting events during ICL repair[24]. However, the And-1 phosphorylation at T826 does not appear to significantly impact HR repair upon DNA double-strand break[23], suggesting distinct regulatory mechanisms governing And-1's role in repairing different types of DNA damage. Moreover, our data also showed that phosphorylation of And-1 not only promotes the association of p125 with gap DNA but also enhances its polymerase activity and processivity (Fig. 5D, E). At this moment, we don't know the detailed molecular mechanism of how And-1 regulates the catalytic activity and processivity of p125. There are two possible mechanisms. First, And-1 may change DNA structure at UV lesion sites to enhance the interaction of p125 with damaged DNA, thus promoting its polymerase activity and processivity. Second, And-1 may simply facilitate

the recruitment of additional p125 molecules to UV lesion sites, increasing the local concentration of polymerase and thereby enhancing both the efficiency and processivity of DNA synthesis. Future structural studies are expected to provide deeper insights into how And-1 and its phosphorylation influence its structural dynamics, DNA affinity, and interactions with other NER repair proteins.

CPDs are the primary drivers of UVB-induced skin tumorigenesis, as they are more mutagenic than 6-4PP in mammalian cells[41–43]. UVB irradiation typically produces far more CPDs than 6-4PPs[44]. Furthermore, CPDs are repaired more slowly than 6-4PPs[45], as corroborated by our results (Fig. 1D–I). While multiple studies have shown that And-1/Ctf4 is an important factor in maintaining genome stability[17,46], its involvement in tumorigenesis remains largely unexplored. Here, our findings demonstrate that And-1 is critical for the efficient removal of CPDs in both cell-based systems (Fig. 3C) and mouse models (Fig. 6C–E), thereby promoting repair synthesis and preserving genomic integrity.

Using And-1 phosphorylation-deficient (*Wdhd1*T819A) mice, we observed the markedly increased susceptibility to keratoacanthoma (KA) (Fig. 7B, C). KA is a fast-growing skin tumor that often resembles cutaneous squamous cell carcinoma (cSCC) but is distinct from it[36,47–49]. Although KA generally exhibits an indolent clinical course and harbors distinct genetic alterations[50], accumulating evidence indicates that a subset of KAs may acquire malignant properties and progress to invasive cSCC under specific conditions[51,52]. Consistent with this, one KA lesion from a *Wdhd1*T819A mice displayed early signs of malignant progression, including mild nuclear atypia and focal basal infiltration, suggestive of early cSCC (Fig. 7Ec). Thus, our in vivo findings underscore the critical role of And-1 in suppressing UVB-induced skin tumorigenesis by maintaining NER efficiency. Notably, And-1 has a mutation rate of 15% (all missense mutations at 8 sites) in cSCC, which is higher than most of the factors involved in NER[53], suggesting a potential link of And-1 with skin cancer. In conclusion, our studies provide the mechanistic base to link And-1 with skin tumorigenesis in mouse models. Future clinical studies may help to investigate the role of And-1 in skin tumorigenesis.

## Methods
### Cell culture

HEK293T cell line was from ATCC (CRL-11268). Immortalized human keratinocyte cell line HaCaT was a gift from Dr. Edward Seto's lab at George Washington University. These two cell lines were cultured in DMEM (Hyclone, SH30243.01) supplemented with 10% FBS. Primary human epidermal keratinocytes (HEKa) (ATCC, PCS-200-011, Lot #81021231) were cultured in Dermal Cell Basal Medium (ATCC, PCS-200-030) supplemented with the Keratinocyte Growth Kit (ATCC, PCS-200-040). Primary human skin fibroblast cell line Hs27 was a gift from Dr. Nady Golestaneh's lab at Georgetown University. Hs27 cells were cultured in DMEM (ATCC, 30-2002) supplemented with 10% FBS. All cells were cultured at 37 °C with 5% $CO_2$ and routinely tested for mycoplasma contamination by PCR. All cells grew to approximately 70–80% confluence prior to UVB irradiation.

### Antibody and plasmids

The p-And-1 was generated as previously described[24]. And-1 antibody was from Novus Biologicals (NBP1-89091, 1:1000). Antibodies against β-Actin (#3700, 1:5000), GAPDH (#2118, 1:5000), FLAG-tag (#14793, 1:1000), V5-tag (#13202, 1:1000), and Histone H3 (#9715, 1:1000) were from Cell Signaling Technology. Antibodies against p125 (sc-17776, 1:1000) and PCNA (sc-56, 1:1000) were from Santa Cruz Biotechnology. Antibody against p66 (#21935-1-AP, 1:1000) was from Proteintech. Antibodies against DDB1(A14242, 1:1000), POLD1/p125 (ab186407, 1:1000), POLD3/p66 (ab182564, 1:1000) were from Abcam. Antibodies against CPDs (CAC-NM-DND-001, 1:1000) and 6-4PPs (CAC-NM-DND-002, 1:1000) were from CosMo Bio. The HRP-conjugated anti-mouse

(#115-035-003, 1:1000) and anti-rabbit (#111-035-144, 1:1000) secondary antibodies were from Jackson ImmunoResearch.

The plasmids containing Full-length And-1 (PEFF-FLAG-And-1), And-1 truncated mutants, and And-1 T826A plasmid were generated as described previously[24]. Plasmid V5-tagged-POLD1-pLX307 was a gift from William Hahn and Sefi Rosenbluh (Addgene plasmid # 98358). Lipofectamine 2000 transfection reagent (Invitrogen, #11668019) was used to transform the plasmids into cells according to the manufacturer's instructions.

### siRNA transfections
The following sequences of siRNA and shRNA duplexes were used in this study: And-1-1, 5'-AGGAAAACAUGCCUGCCAC-3'; And-1-2, 5'-GAAGAUG GUCAAGAAGGCAGCA-3'; si-POLD1/p125, 5'-UUUAAAGAGAGGUUCUUG CTT-3'; and control oligonucleotide (siGL2), AACGUACGCGGAAUACU UCGA; shAnd-1, 5'-AAGCAGGCATCTGCAGCATCC-3'; shSCr, 5'-CAACAA GATGAAGAGCACCAA-3'. The siRNA transfection was performed using Lipofectamine RNAiMAX transfection reagent (Invitrogen, #13778150) for HEK293T, HaCaT, and Hs27 cells. For HEKa cells, transfection was carried out using the Human Keratinocyte Nucleofector® Kit (Lonza, VPD-1002) with the Lonza™ Nucleofector™ 2b Device, according to the manufacturer's instructions.

### Quantification of CPD and 6-4PP by ELISA
Mice epidermal keratinocytes (mEKa) (*Wdhd1* WT and *Wdhd1*[T819A]) or human cells (HEKa, HaCaT and Hs27) transfected with siGL2 or siAnd-1 for 40 h, were exposed to UVB irradiation at 40 mJ/cm², and then collected at 6-, 12-, and 24 h post-UVB treatment. The genomic DNA was immediately extracted using the Quick-DNA™ Miniprep Plus kit (Zymo Research, D4069) and quantified with Eppendorf Biophotometer. DNA samples were converted to single-stranded DNA by incubating at 95 °C for 10 min and then rapidly chilling on ice for 10 min. DNA samples were diluted to 4 µg/mL in cold TE buffer. CPD-DNA and 6-4PP-DNA levels were then measured, respectively, using a commercial ELISA Kit (Cell Biolabs, STA-322-5 and STA-323-5), based on a standard curve constructed from CPD-DNA or 6-4PP-DNA standards provided in the kit, following the manufacturer's instructions.

### Sulforhodamine B (SRB) assay
Cell viability was determined by SRB assay as previously described[54].

### Immunofluorescence (IF) for colocalization
Cells on coverslips were covered with an isoprene polycarbonate membrane filter (pore size, 5 µm; Millipore) and then irradiated with UVB at 75 mJ/cm². The filter was removed, and the cells were cultured in fresh medium for 1 h and then fixed with 4% formaldehyde for 10 min, followed by permeabilization with 0.25% Triton X-100 for 2 min. Cells were then incubated with primary antibodies at 4 °C overnight. After washing three times with PBS, the secondary antibodies, goat anti-Mouse IgG Alexa Fluor™ 488 (0.2 µg/mL, A-10680, Invitrogen) and goat anti-Rabbit IgG Alexa Fluor™594 (2 µg/mL, A-11012, Invitrogen) were incubated at room temperature for 1 h. Nuclear DNA was then stained with DAPI. Images were acquired with an ECHO Revolve Microscope (ECHO, San Diego, CA, USA).

### Unscheduled DNA synthesis (UDS) assay
HaCaT cells were seeded onto 18-mm diameter glass coverslips placed in 6-well plates containing DMEM supplemented with 1% FBS. After 24 h, cells were locally irradiated with UVB at 75 mJ/cm² through a 5-µm pore-size filter. Cells were subsequently pulse-labeled with 20 µM EdU (Jena Bioscience, CLK-N001-100) and 1 µM FuDR (Sigma-Aldrich, F0503) for 2 h. After EdU labeling, cells were then chased with 10 µM thymidine in DMEM without supplements for 30 min and fixed with 3.7% formaldehyde in PBS for 15 min. Cells were then permeabilized

with 0.5% Triton X-100 for 20 min and blocked using 3% BSA. The incorporated EdU was coupled to attoazide Alexa Fluor 488 using Click-iT chemistry according to the manufacturer's instructions. After coupling, cells were fixed with 2% formaldehyde for 10 min and subsequently blocked using 100 mM glycine. DNA was then denatured with 0.5 M NaOH for 5 min on ice, immediately neutralized in 0.1 M Tris-HCl (pH 8.5), and subsequently blocked with 10% BSA for 15 min. Finally, the cells were incubated with CPD antibody (Cosmo Bio, CAC-NM-DND-001, Lot# TM-C-021; 1:1000) for 2 h, followed by incubation with the secondary antibodies for 1 h. Nuclear DNA was stained with DAPI.

### Co-immunoprecipitation (Co-IP)
Cells were lysed on ice for 30 min using a co-IP buffer containing 20 mM Tris-HCl (pH 8.0), 100 mM NaCl, 1 mM EDTA, 0.5% NP-40, 10 mM NaF, and a protease inhibitor cocktail (Thermo Fisher Scientific, A32961). The lysates were then sonicated three times for 10 s/each. After centrifugation, the supernatant was collected and incubated with protein A/G beads coupled with antibodies against the indicated proteins at 4 °C overnight. The beads were washed three times, and the bound proteins were analyzed by Western blotting. For FLAG-IPs, cell lysates were incubated with Anti-FLAG-M2 affinity beads overnight, and the IPs were then subsequently analyzed by Western blotting.

### Mass spectrometry
Mass spectrometry was conducted as previously described[54]. Briefly, HEK293T cells transfected with FLAG-And-1 for 48 h were treated with (*n* = 2) or without (*n* = 2) UVB at 100 mJ/cm² and then harvested 1 h after UVB irradiation. Cell lysates were pulled down using Flag-beads. After extensive washing, immunoprecipitated beads were subjected to proteomics analysis. Proteins were released from beads by using 5% SDS in 50 mM TEAB. Protein samples were incubated with 20 mM DTT (Sigma) at 95 °C for 10 min. After cooling down to room temperature, the solutions were supplemented with 40 mM iodoacetamide and incubated in dark for 30 min. Proteins were then loaded onto S-Trap micro columns (ProtiFi LLC), followed by on-column digestion. The digests were analyzed by nanoUPLC-MS/MS (i.e., nanoAcquity UPLC coupled with an Orbitrap Lumos mass spectrometer) in data-dependent acquisition mode. A 90-min gradient of buffer A (2% ACN, 0.1% formic acid) and buffer B (0.1% formic acid in ACN) was used for peptide separation: 1% buffer B at 0 min, 5% buffer B at 1 min, 22% buffer B at 50 min, 30% buffer B at 65 min, 50% buffer B at 70 min, 98% buffer B at 75 min, 98% buffer B at 90 min. Data were acquired with the Orbitap Fusion Lumos mass spectrometer by using Xcalibur 4.0. The MS raw data files were processed using the MaxQuant software. The human proteome sequences (Uniprot) were used for database search. The false-discovery rate (FDR) was estimated using the fixed value PSM validation. The mass spectrometry data have been deposited to the ProteomeXchange Consortium via the PRIDE partner repository with the dataset identifier PXD052800. Mass spectrometric data are listed in Supplementary Data 1.

### Protein purification and in vitro DNA binding assay
For protein purification, FLAG-tagged human And-1 proteins (wild-type and T826A mutant) and V5-tagged human p125 proteins were purified from HEK293T cells. Plasmids were transfected into HEK293T cells for 48 h, and cells were then exposed to UVB irradiation at 100 mJ/cm² to obtain phosphorylated FLAG-And-1 or left untreated to obtain unphosphorylated FLAG-And-1. One hour after irradiation, cells were harvested and lysed in NETN lysis buffer supplemented with the Pierce Protease and Phosphatase Inhibitor Mini Tablets (Thermo Fisher Scientific, A32961). The lysates were then incubated on ice for 30 min, followed by sonication and centrifugation at 12,000×*g* for 10 min. The supernatants were incubated with ANTI-FLAG® M2 affinity agarose gel (A2220, Millipore Sigma) for FLAG-And-1, or V5-Trap® agarose gel

(v5ta, Proteintech) for V5-p125 at 4 °C overnight. After incubation, the agarose gel was washed three times with NETN lysis buffer, and the bound proteins were eluted using FLAG® Peptide (F3290, Millipore Sigma) for FLAG-And-1, and V5 Peptide (V7754, Sigma-Aldrich) for V5-p125. The eluted And-1 and p125 proteins were concentrated using Ultracel-100K centrifugal filters and stored in small aliquots at −80 °C.

For the in vitro DNA binding assay, the DNA oligos used in this section are listed in Supplementary Data 2. DNA oligos were directly annealed by heating to 95 °C in a water bath for 10 min, followed by slow cooling overnight. The annealed products were then separated on a 10% native PAGE gel at 4 °C. The corresponding gel band of the indicated annealed substrate was excised, purified, and stored at −20 °C. The purified DNA structures were then incubated with purified And-1 or p125 proteins at room temperature for 45 min. Following incubation, streptavidin beads were added, and the mixture was incubated at 4 °C overnight. The beads were then washed three times, and the bound proteins were subjected to SDS-PAGE, followed by immunoblotting with the specified antibodies.

### In vitro DNA polymerase activity assay

The DNA oligos (listed in Supplementary Data 2) were synthesized and purified at Integrated DNA Technologies (IDT). To anneal DNA substrate for primer extension by polymerases: the template oligos (125nt) and FAM-labeled DNA substrate (25nt) are mixed in the annealing buffer containing 10 mM Tris-HCl (pH 7.5), 50 mM NaCl and 0.5 mM EDTA. The oligos mixture was then heated for 5 min at 95 °C and slowly cooled down overnight. The annealed DNA oligos were then incubated with purified V5-p125 (25 nM) together with purified FLAG-And-1 WT or FLAG-And-1 T826A (10 nM, 25 nM, 50 nM) in the reaction buffer, respectively (20 mM Tris-HCL, pH 7.5, 8 mM MgOAc2, 1 mM DTT, 0.1 mg/mL BSA with or without 50−200 nM dNTPs). The reactions were carried out at 37 °C for 30 min and then stopped by the addition of an equal volume of the stop buffer (95% formamide, 25 mM EDTA). Then samples were heated at 100 °C for 15 min to denature the DNA duplex and loaded on a 15% denaturing PAGE to separate the substrate and extended products.

### Generation of Wdhd1$^{T819A}$ mice

All mice were housed in a specific pathogen-free facility under local institutional standards, maintained on a 12-h light/12-h dark cycle at 22 ± 2 °C and 40−60% relative humidity, with free access to standard chow diet and water.

Human And-1 (encoded by WDHD1) T826 locus corresponds to T819 locus in mice. To generate Wdhd1 $^{T819A}$ mice, Wdhd1 T819A point mutation (ACC>GCA) was generated using CRISPR/cas9 gene-editing technology. Briefly, a mixture of guide RNA (gRNA), spCas9 mRNA, and single-stranded oligodeoxynucleotide (ssODN) donor was injected into fertilized C57BL/6J embryos (background strain originally obtained from Jackson Laboratory, USA). Gene editing and embryo preparation were performed by Applied Stem Cell (USA). The gRNA recognition sequence (MC295-3g1R) used in this study was 5′-TTCTTTCTCTTCTTCTGACTGGG-3′. F0 founders were identified by PCR and sequencing analysis (Applied Stem Cell). They were then crossed with 10-week-old wild-type C57BL/6J mice (Jackson Laboratory) to produce F1 pups. To confirm the T819A mutations, mice genotyping and sequencing analysis were performed on purified PCR products, which were amplified using the primers (F): GGTAGTG-CACCCAGACGACTTTCAT and (R): CTTCAACTTGACTTCTGACCCGT GG. All animal studies were approved by the Institutional Animal Care and Use Committee of George Washington University (IACUC Protocol #A2023-059).

### Isolation of murine epidermal keratinocytes

The primary murine epidermal keratinocytes were isolated as previously described[55]. Briefly, both Wdhd1 wild-type (WT) C57BL/6J mice and Wdhd1$^{T819A}$ C57BL/6J mice aged 8 weeks were euthanized by $CO_2$ and the

shaved area of the dorsal skin was treated with depilatory cream. The remaining Nair was rinsed off under running water, and the area was sterilized with Betadine and 70% ethanol. The back skin was dissected, the hypodermis removed, and the tissue cut into small pieces. After three washes with cold PBS, the tissue was flattened on a petri dish with the dermis facing down. Trypsin solution (0.25%) was added, and the skin was allowed to float overnight at 4 °C. Using rounded edge of curved forceps to peel off epidermis and/or scrape off epidermal cells into low Ca$^{++}$ keratinocyte medium (5.4 mM KCl, 0.8 mM $MgSO_4$, 116 mM NaCl, 26 mM $NaHCO_3$, 1 mM $NaH_2PO_4$, 0.1% glucose, 0.001% phenol red, MEM vitamin solution, 4% fetal bovine serum, 10 ng/mL epidermal growth factor and 45 mM $CaCl_2$, pH 7.2). The resulting cell suspension was filtered through a nylon membrane and centrifuged to collect the cell pellet. The isolated cells were passed at least twice before being used in any experiments. Cell confluency was always maintained below 80% to prevent terminal differentiation. The experiments carried out in this section were approved by the Institutional Animal Care and Use Committee of George Washington University (IACUC Protocol #A2023-067).

### UVB irradiation

The UVB irradiation was delivered by a UVB lamp (95-0104-01, UVM-57/ UVP Analytik Jena, 302 nm, 6 watts). The energy output of the UVB lamp was determined by the Transilluminators (Atkinson, NH, U.S.A.) and was revealed to be 1940 μJ/cm$^2$/s at 3 inches from the subjects. Wdhd1 WT C57BL/6J mice and Wdhd1$^{T819A}$ C57BL/6J mice aged 8–10 weeks were anesthetized with isoflurane (Covetrus, 11695067771), and their dorsal skin was shaved. For the single UVB treatment, mice were irradiated at 230 mJ/cm$^2$ and the dorsal skin was collected at 1 h, 8 h, and 24 h after UVB administration. For long-term chronic UVB treatment, mice (6 mice/each group) were irradiated at 180 mJ/cm$^2$ (twice a week) for the first 4 weeks and then were irradiated at 285 mJ/cm$^2$ (thrice a week) for 20 consecutive weeks. Mice were given a one-week break before being sacrificed. Tumor (>1mm$^3$) numbers were counted weekly. Tumor sizes were measured with calipers, and tumor volumes were calculated using the formula: $V = A \times (B^2)/2$, where A and B represent the largest diameter and the smallest diameter of each tumor, respectively. According to the IACUC-approved protocol, the maximal permissible tumor burden was defined as either a single tumor exceeding 250 mm$^3$ or the cumulative volume of multiple tumors on the same mouse exceeding 250 mm$^3$, or the presence of ulceration. Animals reaching this threshold were humanely euthanized, but the tumors were still excised and included in histological and immunohistochemical analyses. Throughout the study, no animal exceeded the maximal tumor size/burden permitted by the ethics committee. Skin tumors or dorsal skin were collected for hematoxylin and eosin (H&E) staining and immunohistochemical (IHC) analysis. The animal study described in this section was approved by the Institutional Animal Care and Use Committees of George Washington University (IACUC Protocol #A2023-067).

### H&E and IHC staining

Mice skin tissues and skin tumors were collected and fixed in 10% neutral-buffered formalin for 36 h, then embedded in paraffin and sectioned at 4-μm thickness for H&E staining and IHC. H&E staining was performed using VitroView™ H&E Stain Kit (VB-3000).

For Ki-67 IHC, deparaffinized sections were immersed in citrate buffer (10 mM sodium citrate, 0.05% Tween-20, pH 6.0) and subjected to antigen retrieval by boiling in a microwave with high power for 3 min and maintained at 95 °C in a steamer for 15 min. After cooling, the sections were blocked with PBS containing normal goat serum for 15 min. Endogenous peroxidase activity was quenched using 1% hydrogen peroxide ($H_2O_2$) in PBS for 15 min. Slides were then incubated overnight at 4 °C with rabbit anti-Ki67 antibody (Abcam, ab15580, 1:1000). The next day, sections were incubated with an HRP-conjugated goat anti-rabbit secondary antibody (Invitrogen, Cat#31460, 1:1000) for 1 h at room temperature. Immunoreactivity

was visualized using DAB solution for 60 s, and nuclei were counter-stained with Mayer's hematoxylin for 60 s.

CPD IHC followed the same antigen retrieval procedure as described above. After cooling, the sections were incubated in 2 M HCl at room temperature for 30 min to denature DNA and then thoroughly washed with PBS. The sections were then blocked with PBS containing normal goat serum for 15 min, followed by additional blocking with the Mouse-on-Mouse (M.O.M.) Kit for 1 h at room temperature. Slides were then incubated overnight at 4 °C with mouse anti-CPD antibody (Cosmo Bio, CAC-NM-DND-001, Lot# TM-C-021; 1:800). After incubation, endogenous peroxidase blocking, secondary antibody incubation, DAB development, and nuclear counterstaining were performed as described for Ki-67.

Quantification of CPD IHC was performed using H-score analysis, focusing exclusively on the epidermal layer. For each time point and group, $n = 6$ mice (independent cohorts) were analyzed. For each mouse, three non-overlapping epidermal fields were randomly selected under a light microscope, with ≥100 epidermal nuclei per field. Scoring was performed blinded to group allocation. Nuclear staining intensity was graded on a four-point scale (0 = negative, 1 = weak, 2 = moderate, 3 = strong brown). For each field, the numbers of cells at each intensity were recorded and converted to percentages, and the H-score was computed as H-score = (% at 1 × 1) + (% at 2 × 2) + (% at 3 × 3), yielding values from 0 to 300. Field-level H-scores were averaged to yield one value per mouse, which served as the unit of analysis for statistics.

### Statistical analysis

Data analysis was conducted using GraphPad Prism 9.0 software. Data are presented as mean ± SD or mean ± SEM, as indicated in the figure legends. Statistical analysis included unpaired two-tailed $t$-test, multiple two-tailed $t$-tests with Holm−Sidak correction, and two-way ANOVA, as specified in figure legends. Statistical significance was considered at $P \leq 0.05$ and denoted as follows: *$P \leq 0.05$, **$P \leq 0.01$, ***$P \leq 0.001$, **$P \leq 0.0001$. Exact $P$-values are indicated in the figures; source data are provided.

### Reporting summary

Further information on research design is available in the Nature Portfolio Reporting Summary linked to this article.

## Data availability

Mass spectrometry data have been deposited in the ProteomeXchange via the PRIDE partner repository and are associated with the accession number PXD052800. All remaining data can be found in the Article, Supplementary, and Source Data files. Source data are provided with this paper.

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

## Acknowledgements

This work was partially supported by funding from the National Institutes of Health (CA247684, CA258357, and CA184717 to W.Z.) and a grant from the McCormick Genomic and Proteomic Center. We extend our gratitude to Dr. Edward Seto from George Washington University for generously providing the HaCaT cell line and to Dr. Nady Golestaneh from Georgetown University for kindly sharing the Hs27 cell line. We sincerely thank Dr. Li Zheng from City of Hope for his valuable advice to optimize the protocol for the DNA polymerase activity assay. Figures 1a, 5a, 6a and 7a are created with BioRender.com, released under a Creative Commons Attribution-NonCommercial-NoDerivs 4.0 International license.

## Author contributions

Shuyan Zhou wrote the manuscript and performed most of the experiments with assistance from Yi Zhang, Zongzhu Li and Penghua Yang. Zhuqing Li assisted with mice genotyping. Patricia S. Latham, Yunxiao Meng and Wen Chen assisted with pathological diagnosis and analysis of the slides. Chunyan Hou and Junfeng Ma designed and performed the mass spectrometry analysis. Wenge Zhu obtained funding to support the project, supervised the project, provided experimental advice, and revised the manuscript.

## Competing interests

The authors declare no competing interests.
