## [Transparent Peer Review file · Nature Communications]

And-1 coordinates with polymerase δ to regulate nucleotide excision repair and UVB-induced skin tumorigenesis

Corresponding Author: Dr Wenge Zhu

Version 0:

Reviewer comments:

Reviewer #1

(Remarks to the Author)

Zhou et al. performed mechanistic studies to demonstrate a critical role of And-1 in nucleotide excision repair (NER) and carcinogenesis. Cell-based studies showed that phosphorylation of And-1 at T826 is essential for its role in NER. And-1 regulates gap-filling DNA synthesis by facilitating the recruitment of p125 to UV lesion sites. And-1 enhances the polymerase activity of p125 in vitro. Mice with And-1 phosphorylation deficiency exhibit increased susceptibility to UVB-induced skin carcinogenesis. The manuscript provides novel molecular insights into how And-1 regulates the later step of NER and how And-1 deficiency augments UV-induced skin carcinogenesis.

The manuscript is well written. The conclusions are mostly supported by the data presented. However, the following major and minor comments should be addressed.

Major comments:

1. Lines 68-70. "dual incisions are produced ... Finally, the gap is filled via DNA synthesis..." -> This sentence is conflicting with the sentence in Line 78 "Once repair synthesis is complete, XPG removes the flap..." Based on the data presented in this manuscript, And-1 deficiency impaired NER. It seems that And-1 is recruited to chromatin after 5' cleavage, but before 3' cleavage occurs. If And-1 is deficient, gap filling will not occur, and 3' cleavage may not occur. That's why UV-induced DNA lesions can be detected in And-1-deficient cells. It will be helpful to elaborate the model for the coordination of dual incision and gap filling.
2. Line 344. "And-1 depletion significantly reduced the co-localization of p66 with CPDs, as well as the association of p66" -> If And-1 is critical for the association of p66 with chromatin, And-1 may interact with p66. However, Figure 2A-C did not include p66. Was p66 not detected in Figure 2A?
3. Line 535. "phosphorylation of And-1 not only promotes the association of p125 with gap DNA, but also enhances p125 polymerase activity (Figure 5D-E)." -> Figure 5E does not show phosphorylation signals of And-1. Also, the Methods section does not describe how "phosphorylated And-1" was prepared for an in vitro DNA polymerase activity assay.

Minor comments:

4. Figure 1A has the label "HEKa cells", but Figure Legend says "(A) Schematic representation of UVB irradiation on HaCaT cells". Which cells are correct?
5. Figure 1B includes the label "10>>r" for horizontal white bars within immunofluorescence images. It may be for "10 μ m", but it is not displayed correctly on the PDF version.
6. The labels for siRNAs in Figures 1D&E do not match with those in Figures F&G. "siAnd-1" should be "siAnd1-1", and "siAnd-2" should be "siAnd1-2".
7. Figure 1D&F. The kinetics of CPD repair seems to be fast, compared to previous knowledge (typical half-life of CPD is ~33 hours). Do you have any explanation for this fast repair?

8. Figure 1F-J Legend and Figure 6D Legend did not include the information on how many experiments were performed to calculate mean +/- SD.

9. Line 310. "p125" -> It will be helpful to say "POLD1/p125" because Figure 2A says "POLD1".

10. Figure 2F Legend is unclear which cell line was used.

11. Figure 3A includes the label "ATRI". Because caffeine targets many enzymes, it is better to label as "caffeine". "ATRI" sounds that an ATR-selective inhibitor was used.

12. Figure 7A shows that the endpoint is 25 weeks after UV. However, Figure 7B-D Legend says the endpoint is "week 24". Should this be "week 25"? Figure 7C graph shows data points at "week 25".

Reviewer #2

(Remarks to the Author)

In this manuscript, the authors describe that And-1 localizes at UV damage sites and helps the repair of UV-induced DNA lesions. They identified DNA Poldelta-p125 as an And-1 binding protein, and the And-1 interaction with p125 is stimulated by UV exposure. And-1 promotes p125 recruitment to UV damage to facilitate the repair of UV damage. They found that And-1 stimulates the polymerase activity of p125 at gap substrates. Furthermore, And-1 phosphorylation at T826 by ATR, which previously has been shown to be important for ICL repair by the same group, also promotes And-1 interaction with UV-damaged DNA and p125. They then generated the phosphor-deficient And-1 knockin mice and showed that these mice exhibited impaired DNA repair and increased formation of keratoacanthomas under chronic UVB exposure. These results suggest that And-1 is a critical component of the NER pathway by promoting PolDelta activity and may suppress keratoacanthomas. The manuscript is well written, and overall the results are clear and support the major conclusions. Findings provide novel insights into the NER pathway.

Comments:

1. Throughout the figures (Fig. 2, 3, 4, 6, chromatin fractions were prepared to show protein binding/recruitment to chromatin. However, while Histone H3 is a good control for chromatin fraction, GAPDH blots should be included to show that the isolated chromatin fraction is free of cytoplasmic protein contamination.
2. Fig. 3 and Supplementary Fig. 3: Caffeine is not a specific inhibitor for ATR. It also inhibits ATM, mTOR, DNAPK, and other kinases. A more specific ATR inhibitor like VE821 or siATR should be used.
3. Keratoacanthoma (KA) is a benign skin tumor. Is there any clinical evidence associating And-1 with KA? Also, did the Knock-in mice develop other types of skin tumor such as melanoma or SCC? In pictures shown in Fig. 7B, it appears that the knockin mice developed more pigmentation and more dark spots, but it is difficult to tell whether melanoma was developed. Can authors stain the tissues with markers for KA and melanoma?
4. Page 15, Paragraph 2: It would be more helpful if the authors can elaborate on what WD40, SepB and HMG domains are and their molecular functions. Are they involved in DNA binding? Which domain is involved in protein-protein interaction?
5. Fig. 5A. The gel showing the purity of 4 different DNA substrates should be provided.
6. Fig. 5E. It looks like And-1 phosphorylation promotes the processivity of Poldelta. The authors may consider discussing this in the text.

Reviewer #3

(Remarks to the Author)

AND-1 is a highly conserved component of the replication progression complex. AND-1 regulates DNA replication and bridges the CMG helicase and DNA polymerase alpha. Its acute loss causes the accumulation of cells at the G2 phase, and the occurrence of chromosome breaks in mitotic chromosome spreads. The authors here uncovered the role of AND-1 in nucleotide excision repair (NER). They measured the repair kinetics of UV lesions (6-4PP and CPD) in the HaCaT keratinocyte line (Fig. 1), the physical association of AID-1 and Pol-delta (Fig. 2), the role of ATR-mediated phosphorylation of AND-1 at T826 (Fig. 3), the phosphorylation of AND-1 controls its recruitment to UV-irradiated sites (Fig. 4), wild-type AND-1, but not T826A mutant, promotes in vitro DNA synthesis by the p125 subunit of Pol-delta holoenzyme (Fig. 5), the T826A mutation delays the UV damage repair kinetics of skin cells of mice (Fig. 6), and the T826A mutation increased the UV-induced skin cancer (Fig. 7). The authors conducted an impressive amount of work, using biochemical methods, sophisticated cellular analyses, and mice. They proposed a surprising conclusion: the role of AND-1 in NER. However, the reviewer feels that the work does not fully support the conclusions (see specific comments) and could not support the publication of this manuscript in Nature Commun.

Specific comments.

1. The authors did not accurately measure the repair kinetics of two UV photoproducts, CPDs, and 6-4PPs, after UV irradiation. Figures 1 and 6 indicate that wild-type (WT) cells repaired CPDs and 6-4PPs with similar kinetics. WT cells repaired 80-90% of CPDs and 90-95% of 6-4PPs at 24 h after UV irradiation in the keratinocyte line (Fig.1F and G) and murine primary skin cells (Fig.6D and F). However, such similar repair kinetics of CPDs and 6-4PPs do not agree with the previously published data, including Fig.2 in Nature 1995 (PMID 7675085), Fig.2 in Mol Cell 2000 (PMID: 10882109), Fig.3 and S3 in Mol Cell 2023 (PMID: 37816354). These studies showed that the repair kinetics of CPDs is more than three times slower than that of 6-4PPs in mouse primary skin fibroblasts (Nature 1995), hamster cells (Mol Cell 2000), and human RPE cells (Mol Cell 2023). The authors must measure the repair kinetics of UV lesions again using another method, such as

ELISA.

2. Comment related with 1. The authors need to show the calibration curve by exposing cells to 2, 4, 6, 8, and 10 J/m², immediately purifying genomic DNA, and measuring the number of CPDs and 6-4PPs.
3. Comment related with 1. WT cells repaired 50-60% of CPDs and 6-4PPs up to 1 h and subsequently repaired only around 20% for 16 hours (between 8 and 24 h) after UV irradiation (Fig.1F and G). Such a significant difference in the repair kinetics was not seen in the previous publications. The authors must measure the repair kinetics of UV lesions again
4. Comment related with 1. The repair kinetics of CPDs and 6-4PPs were much faster in WT cells than siAND-1 treated cells from 0 to 1 hour post-UV irradiation. Then, the repair kinetics of CPDs and 6-4PPs were significantly slower in WT cells than siAND-1 treated cells from 1 to 8 hours post-UV irradiation, with these two cell populations repairing UV lesions at the same kinetics. The prominent contribution of AND-1 to UV damage repair only up to 1-hour post-UV irradiation does not agree with the authors' conclusion, 'with the maximum accumulation of And-1 at 4 hours and p125 at 8 hours after UVB exposure (Figure 2D), suggesting that And-1 may be involved in NER late stage' (line 320, page 12).
5. In Fig. 2D, AND-1 and p125 significantly accumulated in the chromatin fraction from 1 to 8 h after UV irradiation, but at that time, AND-1 did not promote the repair of UV damage according to Fig.1F, G. Thus, contrary to UV damage repair only up to 1-hour post-UV irradiation disagrees with the authors' conclusion, the recruitment of AND-1 to UV-damaged sites (Fig.2D) does not cause the AND-1-mediated promotion of UV damage repair.
6. The authors analyzed the recruitment of p125 to UV-damaged sites, comparing WT cells and siAND-1 treated cells in Figures 2, 3C, E, F, and 4B. To examine the functionality of Pol-delta more accurately, the authors must measure unscheduled DNA synthesis (UDS), which measures DNA synthesis during NER and is the standard method for diagnosing NER-deficient XP patients.
7. Previous studies have established the in vitro reconstitution of all NER reactions using DNA carrying UV damage. The authors need to analyze the effect of AND-1 on this in vitro reconstitution reaction. Alternatively, they must examine the impact of AND-1 on NER by measuring UDS.
8. The authors concluded that ATR-mediated phosphorylation of AND-1 at T826 promotes NER by showing the reduced functioning of the AND-1 T826A mutant protein. The authors have to exclude the possibility that the T826A mutation interferes with the functioning of the AND-1 independent of ATR. To this end, the authors need to show that treatment of WT cells with ATR inhibitor (ATRi) and treatment of cells expressing T826A mutant protein with ATRi delay UV damage repair and UDS to the same extent. The authors also must show that the T826E phosphomimetic mutation reverses the inhibitory effect of ATRi on NER at least partially.
9. The latency period of skin carcinogenesis was amazingly short (multiple tumor formation in the analyzed mice at 25 weeks of irradiation in Fig. 7C) despite the modest defect in UV damage repair in mice carrying the T826A mutation (Fig. 6D, F). This latency period is comparable to previously published latency periods of UV-irradiated XPA-deficient mice, which have a considerably more severe defect in UV damage repair (no repair) in skin tissue than skin cells carrying the T826A mutation (Fig. 6D, F). The short latency period suggests that the T826A mutation impairs the functionality of AND-1, independent of its potential role in NER. Another possibility is that the authors may have counted all skin tumors without distinguishing malignant tumors from benign ones.
10. Related to comment 9. Previous studies indicate that the latency period of skin cancer in UV-irradiated WT mice is >30 weeks of irradiation, whereas the authors showed that skin cancer arose at 25 weeks of irradiation (Fig.7C). The latency period of skin carcinogenesis might be too short.

Minor comments

It is unclear whether the skin tumors in Fig.2E are benign or malignant. The authors need to present magnified histological images that clearly show evidence of malignancy (skin tumors growing through the basement membrane and into the mesenchymal layer).

Version 1:

Reviewer comments:

Reviewer #1

(Remarks to the Author)

Thank you for thoroughly revising the manuscript. The authors fully addressed reviewers' comments. The revised manuscript is suitable for publication.

Reviewer #2

(Remarks to the Author)

The revised manuscript has addressed my concerns and additional data significantly strengthen the manuscript.

Reviewer #3

(Remarks to the Author)

The authors have addressed all the concerns the reviewer raised. The reviewer would like to support the publication of their manuscript in Nature Communications.

Minor comments:

(1) The authors need to tone down the description of keratoacanthoma (KA): 'KA is currently classified as a low-grade variant of cutaneous squamous cell carcinoma (cSCC)' (Line 662-663). The website of NIH/NCI

(<https://www.cancer.gov/publications/dictionaries/cancer-terms/def/keratoacanthoma>) shows that 'They may look like squamous cell skin cancer, but they are usually benign and rarely spread to other parts of the body' but not cSCC. In the abstract of a recent review (PMID: 39449378), KA is defined as benign tumors (KAs are rapidly growing, benign squamous tumors that are typically well differentiated) but not cSCC.

(2) The authors need to cite PMID: 39449378 on lines 581 and 664.

REVIEWER COMMENTS

Reviewer #1 (Remarks to the Author):

Zhou et al. performed mechanistic studies to demonstrate a critical role of And-1 in nucleotide excision repair (NER) and carcinogenesis. Cell-based studies showed that phosphorylation of And-1 at T826 is essential for its role in NER. And-1 regulates gap-filling DNA synthesis by facilitating the recruitment of p125 to UV lesion sites. And-1 enhances the polymerase activity of p125 in vitro. Mice with And-1 phosphorylation deficiency exhibit increased susceptibility to UVB-induced skin carcinogenesis. The manuscript provides novel molecular insights into how And-1 regulates the later step of NER and how And-1 deficiency augments UV-induced skin carcinogenesis.

The manuscript is well written. The conclusions are mostly supported by the data presented. However, the following major and minor comments should be addressed.

We thank the reviewer for overall positive comments on our manuscript. We have addressed all concerns as below.

Major comments:

1. Lines 68-70. “dual incisions are produced ... Finally, the gap is filled via DNA synthesis...” -> This sentence is conflicting with the sentence in Line 78 “Once repair synthesis is complete, XPG removes the flap...” Based on the data presented in this manuscript, And-1 deficiency impaired NER. It seems that And-1 is recruited to chromatin after 5’ cleavage, but before 3’ cleavage occurs. If And-1 is deficient, gap filling will not occur, and 3’ cleavage may not occur. That’s why UV-induced DNA lesions can be detected in And-1-deficient cells. It will be helpful to elaborate the model for the coordination of dual incision and gap filling.

We thank the reviewer for this insightful comment. To address this comment, we have revised the Introduction to eliminate any conflict descriptions about the timing of incision and gap-filling steps (Page3: Lines70-72, Lines79-81). In addition, in the Discussion, we have elaborate a model for the coordination of incision and gap filling synthesis, as well as potential role of And-1 in NER (Page 23, lines 616-624).

2. Line 344. “And-1 depletion significantly reduced the co-localization of p66 with CPDs, as well as the association of p66” -> If And-1 is critical for the association of p66 with chromatin, And-1 may interact with p66. However, Figure 2A-C did not include p66. Was p66 not detected in Figure 2A?

Thanks for this comment. The reason that we did not examined the intraction of And-1 with p66 is because p66 was not detected in mass spectrometry analysis shown in Figure 2A and the goal of Figure 2B is to confirm the interaction of And-1 with NER factors identified from mass spectrometry analysis shown in Figure 2A. As shown in new Supplementary Figure 2F, co-immunoprecipitation assay indicated that p66 was detected from FLAG–And-1 IPs and the interaction between And-1 and p66 was increased in cells treated with UVB.

3. Line 535. “phosphorylation of And-1 not only promotes the association of p125 with gap DNA, but also enhances p125 polymerase activity (Figure 5D-E).” -> Figure 5E does not show phosphorylation signals of And-1. Also, the Methods section does not describe how “phosphorylated And-1” was prepared for an in vitro DNA polymerase activity assay.

*Thanks for the comment. In revision, we have examined the phosphorylation of And-1 and the new data were now included in **Figure 5E**. Additionally, in revised Methods section, we have described how phosphorylated And-1 was prepared (**Page 8, lines 214-215**).*

Minor comments:

4. Figure 1A has the label “HEKa cells”, but Figure Legend says “(A) Schematic representation of UVB irradiation on HaCaT cells”. Which cells are correct?

Thanks for the comment. We apologize for the mistake. The correct cell type used in Figure 1A is HEKa cells. We have corrected it in revision.

5. Figure 1B includes the label “10>>1” for horizontal white bars within immunofluorescence images. It may be for “10 μm”, but it is not displayed correctly on the PDF version.

Thanks for the comment. We have corrected it as suggested in revision.

6. The labels for siRNAs in Figures 1D&E do not match with those in Figures F&G. “siAnd-1” should be “siAnd1-1”, and “siAnd-2” should be “siAnd1-2”.

Thanks for the comment. We apologize for the mistake. We have corrected them as suggested in revision.

7. Figure 1D&F. The kinetics of CPD repair seems to be fast, compared to previous knowledge (typical half-life of CPD is ~33 hours). Do you have any explanation for this fast repair?

*We thank the reviewer for the comment. We agree with reviewer that the CPD repair kinetics shown in the Figure 1D and 1F of the first submission is faster than what has been reported. This observation is most likely due to the use of dot blot assays to measure CPD in the first version of manuscript. Dot blot assay to measure CPD are generally less sensitive and less quantitative than ELISA-based detection method. It was also suggested by reviewer 3 that the kinetics of UVB-induced DNA lesion repair should be measured by using a more sensitive ELISA-based assay. To address these concerns, as suggested, we used ELISA-based assay to measure CPD levels in three different cell lines: HEKa, HaCaT, and Hs27. The results showed that the remaining CPD levels at 24 hours post-UVB exposure (40 mJ/cm²) were approximately 38%, 52%, and 57%, respectively, which are consistent with previous reports (PMID: 7675085, PMID: 10882109, PMID: 37816354). In revision, we have replaced dot blot data with ELISA-based data in **Figure 1D–I; Figure 3C, I; Figure 6C**.*

8. Figure 1F-J Legend and Figure 6D Legend did not include the information on how many experiments were performed to calculate mean +/- SD.

*Thank you for the comment. We have revised the figure legends for **Figures 1 and 6** to clarify that the ELISA data are presented as mean \pm SEM from three independent experiments.*

9. Line 310. “p125” -> It will be helpful to say “POLD1/p125” because Figure 2A says “POLD1”.

Thanks for the comment. We have revised it as suggested.

10. Figure 2F Legend is unclear which cell line was used.

*Thank you for the comment. The cell line used in **Figure 2F** is HaCaT, and we have revised the figure legend accordingly.*

11. Figure 3A includes the label “ATRi”. Because caffeine targets many enzymes, it is better to label as “caffeine”. “ATRi” sounds that an ATR-selective inhibitor was used.

*Thanks for the comment. Since reviewer 2 also suggested to use ATR specific inhibitor, we therefore repeated the experiments using VE-821, a selective ATR inhibitor. The updated data with VE-821 are now replaced the original caffeine-based results, and new data are included in **Figures 3A and Figure 4D** in revision.*

12. Figure 7A shows that the endpoint is 25 weeks after UV. However, Figure 7B-D Legend says the endpoint is “week 24”. Should this be “week 25”? Figure 7C graph shows data points at “week 25”.

*We thank the reviewer for the comment. The correct endpoint for the experiment is week 25 after UV exposure. We have corrected the legend in **Figures 7B–D**.*

Reviewer #2 (Remarks to the Author):

In this manuscript, the authors describe that And-1 localizes at UV damage sites and helps the repair of UV-induced DNA lesions. They identified DNA Poldelta-p125 as an And-1 binding protein, and the And-1 interaction with p125 is stimulated by UV exposure. And-1 promotes p125 recruitment to UV damage to facilitate the repair of UV damage. They found that And-1 stimulates the polymerase activity of p125 at gap substrates. Furthermore, And-1 phosphorylation at T826 by ATR, which previously has been shown to be important for ICL repair by the same group, also promotes And-1 interaction with UV-damaged DNA and p125. They then generated the phosphor-deficient And-1 knockin mice and showed that these mice exhibited impaired DNA repair and increased formation of keratoacanthomas under chronic UVB exposure. These results suggest that And-1 is a critical component of the NER pathway by promoting PolDelta activity and may suppress keratoacanthomas. The manuscript is well written, and overall the results are clear and support the major conclusions. Findings provide novel insights into the NER pathway.

We thank the reviewer for overall positive comments on our manuscript. We have addressed all concerns as below.

Comments:

1. Throughout the figures (Fig. 2, 3, 4, 6, chromatin fractions were prepared to show protein binding/recruitment to chromatin. However, while Histone H3 is a good control for chromatin fraction, GAPDH blots should be included to show that the isolated chromatin fraction is free of cytoplasmic protein contamination.

*We appreciate the reviewer's valuable suggestion. As recommended, we have included GAPDH as a negative control to confirm the absence of cytoplasmic contamination in the chromatin fractions in **Figure 2D,2E, Supplementary Figure 2D,2E,2I-2K, Figure 3D, Figure 4C and Figure 6B.***

2. Fig. 3 and Supplementary Fig. 3: Caffeine is not a specific inhibitor for ATR. It also inhibits ATM, mTOR, DNAPK, and other kinases. A more specific ATR inhibitor like VE821 or siATR should be used.

*Thank you for the comment. As suggested, we have conducted experiments using the ATR-specific inhibitor VE-821 and the new data were included in **Figure 3A.** We have also repeated the experiment in **Figure 4D** using VE-821 accordingly.*

3. Keratoacanthoma (KA) is a benign skin tumor. Is there any clinical evidence associating And-1 with KA? Also, did the Knock-in mice develop other types of skin tumor such as melanoma or SCC? In pictures shown in Fig. 7B, it appears that the knockin mice developed more pigmentation and more dark spots, but it is difficult to tell whether melanoma was developed. Can authors stain the tissues with markers for KA and melanoma?

We thank the reviewer for the thoughtful comments.

*To our knowledge, there is no direct clinical evidence linking And-1 to keratoacanthoma (KA) development in human patients. At present, there is no definitive immunohistochemical marker that can reliably distinguish KA from cutaneous squamous cell carcinoma (SCC). Therefore, the clinical diagnosis is primarily based on a combination of histomorphological features and Ki-67 immunostaining. In particular, the presence or absence of basal infiltration serves as a key histological criterion, while Ki-67 staining provides valuable insight into the proliferative activity of the lesion. Together, these factors aid in distinguishing benign from malignant processes. In our study, all tumors collected from both WT and *Wdhd1*^{T819} mice were independently reviewed by three experienced pathologists. All lesions were diagnosed as keratoacanthomas (KAs) at various stages of progression (See updated **Supplementary Figure S7F1-K2, Supplementary Table 3**). Only one tumor—from *Wdhd1*^{T819A} mouse #3—exhibited features of early SCC, including mild nuclear atypia and focal basal infiltration (**Figure 7Ec, Supplementary Figure SH1-2**).*

*Thanks for the comments about increased pigmentation in *Wdhd1*^{T819} mice (Fig. 7B). To address this question, we performed immunohistochemical staining (using AP/Fast Red detection) for SOX10, Melan-A, and Ki-67 on the available *Wdhd1*^{T819A} mice tumor samples on the pigmented areas. As shown in **Supplementary Figure 8A-E**, the stained sections from the pigmented areas demonstrated the focal positive expression of SOX10, rare expression of Melan-A, and extremely low Ki-67 proliferative activity. The nuclei exhibited normal morphology without atypia. Based on these histopathological and immunohistochemical findings, the pigmentation was diagnosed as a benign post-inflammatory or UV-induced melanotic pigmentation, rather than indicative of melanoma.*

4. Page 15, Paragraph 2: It would be more helpful if the authors can elaborate on what WD40, SepB and HMG domains are and their molecular functions. Are they involved in DNA binding? Which domain is involved in protein-protein interaction?

*We thank the reviewer for this suggestion. We have now added a brief introduction of the three domains of And-1 and their respective molecular functions to the revised manuscript (**Page 4, Paragraph 2**). Briefly: And-1 is characterized by three major domains: an N-terminal WD40 repeat domain, consists of conserved WD repeat motifs that form a β -propeller structure, typically acting as protein interaction scaffolds; a central SepB domain that specifically mediates protein-protein interactions; and a C-terminal HMG (High Mobility Group) domain that facilitates chromatin engagement through DNA binding.*

5. Fig. 5A. The gel showing the purity of 4 different DNA substrates should be provided.

*Thank you for the comment. We have performed the experiment as suggested. The new data are now included in **Supplementary Figure 5A**.*

6. Fig. 5E. It looks like And-1 phosphorylation promotes the processivity of Poldelta. The authors may consider discussing this in the text.

*We appreciate the reviewer's suggestion. We have discussed it in our manuscript as suggested (**Page 24, lines 644-650**).*

Reviewer #3 (Remarks to the Author):

AND-1 is a highly conserved component of the replication progression complex. AND-1 regulates DNA replication and bridges the CMG helicase and DNA polymerase alpha. Its acute loss causes the accumulation of cells at the G2 phase, and the occurrence of chromosome breaks in mitotic chromosome spreads. The authors here uncovered the role of AND-1 in nucleotide excision repair (NER). They measured the repair kinetics of UV lesions (6-4PP and CPD) in the HaCaT keratinocyte line (Fig. 1), the physical association of AID-1 and Pol-delta (Fig. 2), the role of ATR-mediated phosphorylation of AND-1 at T826 (Fig. 3), the phosphorylation of AND-1 controls its recruitment to UV-irradiated sites (Fig. 4), wild-type AND-1, but not T826A mutant, promotes in vitro DNA synthesis by the p125 subunit of Pol-delta holoenzyme (Fig. 5), the T826A mutation delays the UV damage repair kinetics of skin cells of mice (Fig. 6), and the T826A mutation increased the UV-induced skin cancer (Fig. 7). The authors conducted an impressive amount of work, using biochemical methods, sophisticated cellular analyses, and mice. They proposed a surprising conclusion: the role of AND-I in NER. However, the reviewer feels that the work does not fully support the conclusions (see specific comments) and could not support the publication of this manuscript in Nature Commun.

We thank the reviewer for the thorough and constructive comments on our manuscript. We have addressed all concerns as below.

Specific comments.

1. The authors did not accurately measure the repair kinetics of two UV photoproducts, CPDs, and 6-4PPs, after UV irradiation. Figures 1 and 6 indicate that wild-type (WT) cells repaired CPDs and 6-4PPs with similar kinetics. WT cells repaired 80-90% of CPDs and 90-95% of 6-4PPs at 24 h after UV irradiation in the keratinocyte line (Fig.1F and G) and murine primary skin cells (Fig.6D and F). However, such similar repair kinetics of CPDs and 6-4PPs do not agree with the previously published data, including Fig.2 in Nature 1995 (PMID 7675085), Fig.2 in Mol Cell 2000 (PMID: 10882109), Fig.3 and S3 in Mol Cell 2023 (PMID: 37816354). These studies showed that the repair kinetics of CPDs is more than three times slower than that of 6-4PPs in mouse primary skin fibroblasts (Nature 1995), hamster cells (Mol Cell 2000), and human RPE cells (Mol Cell 2023). The authors must measure the repair kinetics of UV lesions again using another method, such as ELISA.

We sincerely thank the reviewer for the insightful comment and suggestion to use ELISA to measure the repair kinetics of UV lesions. We agree with the reviewer that our initial data, which were based on

immunoblotting, appeared to show similar repair kinetics for CPDs and 6-4PPs, which is inconsistent with the previous observation obtained by ELISA-based assays.

To address these concerns, as suggested by the reviewer, we re-evaluated the repair kinetics of UV-induced DNA lesions using a commercially available ELISA kit specific for CPDs and 6-4PPs. These assays were performed in three different cell lines (HEK293, HaCaT, Hs27) by following the manufacturer's instructions. The new data are now included in the revised **Figure 1D-I**. Moreover, we re-assessed photolesion levels in murine primary keratinocytes following UVB exposure using the same ELISA-based method. These additional new data are now incorporated into the revised **Figure 6C**.

We believe the difference between the initial blot-based results and the new ELISA data arises from the limited sensitivity and semi-quantitative nature of Western blotting, while, ELISA offers greater sensitivity and quantitative accuracy. Accordingly, we have replaced the original immunoblot data with the ELISA-based results in the revised manuscript (**Figure 1D-I; Figure 3C, I; Figure 6C**). Again, we are grateful for the reviewer's suggestion, which strengthens the rigor of our study.

2. Comment related with 1. The authors need to show the calibration curve by exposing cells to 2, 4, 6, 8, and 10 J/m², immediately purifying genomic DNA, and measuring the number of CPDs and 6-4PPs.

We thank the reviewer for the valuable suggestion. In our revised experiments, as suggested, we employed a commercially available CPD/6-4PP ELISA kit (**Cell Biolabs, STA-322-5 and STA-323-5**) that includes certified DNA standards with known quantities of CPD and 6-4PP lesions. Using this approach, we generated a standard calibration curve to quantify photoproduct levels in our samples (**Rebuttal Figure 1**).

We believe that the use of these well-characterized and commercially validated standards provides a reliable and standardized approach for accurate photoproduct quantification. We hope this approach addresses the reviewer's concern.

Rebuttal Figure 1. Left panel, the CPD-DNA standard curve by ELISA assay. Right panel, the 6-4PP-DNA standard curve by ELISA assay.

3. Comment related with 1. WT cells repaired 50-60% of CPDs and 6-4PPs up to 1 h and subsequently repaired only around 20% for 16 hours (between 8 and 24 h) after UV irradiation (Fig.1F and G). Such a significant difference in the repair kinetics was not seen in the previous publications. The authors must measure the repair kinetics of UV lesions again.

Thanks for the comments. Please also see our response in questions #1 and #2. The new data obtained using ELISA kit (Figure 1D-I) indicated that a gradual and lesion-specific repair process: 6-4PPs are removed more efficiently than CPDs, with approximately 85-90% of 6-4PPs and 40-60% of CPDs (depending on the cell line) repaired by 24 hours following UVB exposure at 40 mJ/cm². These findings are consistent with previously established observations that 6-4PPs are repaired significantly faster than CPDs.

4. Comment related with 1. The repair kinetics of CPDs and 6-4PPs were much faster in WT cells than siAND-1 treated cells from 0 to 1 hour post-UV irradiation. Then, the repair kinetics of CPDs and 6-4PPs were significantly slower in WT cells than siAND-1 treated cells from 1 to 8 hours post-UV irradiation, with these two cell populations repairing UV lesions at the same kinetics. The prominent contribution of AND-1 to UV damage repair only up to 1-hour post-UV irradiation does not agree with the authors' conclusion, 'with the maximum accumulation of And-1 at 4 hours and p125 at 8 hours after UVB exposure (Figure 2D), suggesting that And-1 may be involved in NER late stage' (line 320, page 12).

Thanks for this comments. We realized the limitation and weakness of immunoblot-based assay to measure CPDs and 6-4PPs as we addressed above. Therefore, in revision, we repeated these experiments using ELISA-based assay and the new data are now included in Figure 1D-I. As shown in Figure 1D-I, WT cells (HEKa, HaCaT, Hs27) exhibit robust, progressive removal of CPDs, particularly between 6 and 24 hours, whereas siAnd-1-treated cells show a significant delay of lesion removal over the same period. These dynamics are consistent with chromatin recruitment results shown in Figure 2D, in which And-1 and p125 peaks after 4 hours post-UVB exposure. Together, these findings support our conclusion that And-1 functions primarily in the late stage of NER, particularly in gap-filling repair synthesis.

5. In Fig. 2D, AND-1 and p125 significantly accumulated in the chromatin fraction from 1 to 8 h after UV irradiation, but at that time, AND-1 did not promote the repair of UV damage according to Fig.1F, G. Thus, contrary to UV damage repair only up to 1-hour post-UV irradiation disagrees with the authors' conclusion, the recruitment of AND-1 to UV-damaged sites (Fig.2D) does not cause the AND-1-mediated promotion of UV damage repair.

Thanks for the comments. We have addressed these concerns in #4. Again, the new ELISA data (Figure 1D-I) clearly demonstrate that CPD and 6-4PP repair in siAnd-1-treated cells was impaired at 6-24 hours post-UVB exposure, compared to WT cells. This delayed repair is consistent with chromatin fraction data (Figure 2D), which show maximum And-1 accumulation of And-1 and p125 after 4 hours post-irradiation.

6. The authors analyzed the recruitment of p125 to UV-damaged sites, comparing WT cells and siAND-1 treated cells in Figures 2, 3C, E, F, and 4B. To examine the functionality of Pol-delta more accurately, the authors must measure unscheduled DNA synthesis (UDS), which measures DNA synthesis during NER and is the standard method for diagnosing NER-deficient XP patients.

We addressed both comments #6 and #7 together in #7.

7. Previous studies have established the in vitro reconstitution of all NER reactions using DNA carrying UV damage. The authors need to analyze the effect of AND-1 on this in vitro reconstitution reaction. Alternatively, they must examine the impact of AND-1 on NER by measuring UDS.

Thanks for these comments.

*As suggested, we performed **unscheduled DNA synthesis (UDS) assays** by measuring EdU incorporation following UVB irradiation—an well-established approach to assess NER-dependent repair synthesis in suggested figures. These new data are now included in **Figure 2H, Figure 3F-H, Supplementary Figure 3I-K, and Supplementary Figures 4A–B**. These new data clearly demonstrate that cells with And-1–deficiency including mutations showed a significant reduction in UV-induced EdU incorporation, indicating a critical role of And-1 in NER-associated repair synthesis.*

The UDS assay provides direct and robust in vitro evidence strongly demonstrating that And-1 is required for Pol δ –mediated repair synthesis in NER. Thus, our study have provided both cell-based and animal data to desomtrate the critical role of And-1 in NER. Although the in vitro reconstitution of NER with purified proteins have been established, we acknowledge that these in vitro assay will take significant amount of time and may not be able to reflect the reality of mechanism in cells and in animal. And yet, it is our future plan to using this in vitro assay to elucidate the molecular detail of how And-1 regulates NER.

8. The authors concluded that ATR-mediated phosphorylation of AND-1 at T826 promotes NER by showing the reduced functioning of the AND-1 T826A mutant protein. The authors have to exclude the possibility that the T826A mutation interferes with the functioning of the AND-1 independent of ATR. To this end, the authors need to show that treatment of WT cells with ATR inhibitor (ATRi) and treatment of cells expressing T826A mutant protein with ATRi delay UV damage repair and UDS to the same extent. The authors also must show that the T826E phosphomimetic mutation reverses the inhibitory effect of ATRi on NER at least partially.

We thank the reviewer for the insightful comment. As suggested, we conducted experiments by comparing the UVB-induced CPD removal and UDS in following cells: wild-type (shScramble) HaCaT cells \pm ATR

*inhibitor (VE-821), and And-1–depleted cells reconstituted with empty vector, T826A, or T826E ± ATRi. These new data are now presented in **Figure 3I and Supplementary Figure S3I–K.***

9. The latency period of skin carcinogenesis was amazingly short (multiple tumor formation in the analyzed mice at 25 weeks of irradiation in Fig. 7C) despite the modest defect in UV damage repair in mice carrying the T826A mutation (Fig. 6D, F). This latency period is comparable to previously published latency periods of UV-irradiated XPA-deficient mice, which have a considerably more severe defect in UV damage repair (no repair) in skin tissue than skin cells carrying the T826A mutation (Fig. 6D, F). The short latency period suggests that the T826A mutation impairs the functionality of AND-1, independent of its potential role in NER. Another possibility is that the authors may have counted all skin tumors without distinguishing malignant tumors from benign ones.

We thank the reviewer for this insightful comment.

We agree that the relatively short latency period observed in $Wdhd1^{T819A}$ mice compared to the XPA-deficient mice in the previous study. We believe that this difference is mainly due to the following reasons: (1) All tumor lesions observed in $Wdhd1^{T819A}$ mice were confirmed as keratoacanthomas (KAs) with different stages of progression, but not cutaneous squamous cell carcinomas (cSCCs), while XPA-deficient mice developed cSCCs. As we know, KAs are known for their rapid appearance—typically developing within a few weeks to months of UV exposure—and often regress spontaneously, compared to cSCCs, which progress more slowly (See book by Patrick M. Zito and Richard Scharf. Keratoacanthoma. Treasure Island (FL): StatPearls). Clinically and pathologically, KA is recognized as a distinct and often earlier-arising lesion compared to SCC, particularly in the context of chronic UV exposure. (2) AND-1 (also known as Ctf4) plays multifaceted roles in genome maintenance, including replication fork stability, coordination of DNA replication and repair. These multifaceted roles may lead to the short latency period of skin keratoacanthoma (not cSCCs).

10. Related to comment 9. Previous studies indicate that the latency period of skin cancer in UV-irradiated WT mice is >30 weeks of irradiation, whereas the authors showed that skin cancer arose at 25 weeks of irradiation (Fig.7C). The latency period of skin carcinogenesis might be too short.

We appreciate the reviewer’s careful assessment and thoughtful observation.

We believe that the difference from previous reports is due to the following reasons: (1) The sole tumor lesion observed in our WT group at 25 weeks of irradiation was confirmed as a keratoacanthoma (KA) at an early proliferating stage, rather than a cutaneous squamous cell carcinoma (cSCC) that was reported

in previous study (PMID: 7675085). KAs are generally considered low-grade variants of cSCC and can arise earlier under chronic UV stress.

(2) Differences in latency periods can be influenced by several experimental variables, including the genetic background of the mice, UV spectrum, UV dosage, frequency of irradiation, and environmental conditions. Although our UVB treatment schedule was shorter in total duration than that reported in previous studies, it consisted of 180 mJ/cm² twice per week for 4 weeks, followed by 285 mJ/cm² three times per week for 20 weeks. Compared to a previous study (PMID: 7675085) that used 200 mJ/cm² three times weekly for 39 weeks and did not observe tumor formation in WT mice, our regimen applied a higher dose per session during the later phase and delivered the cumulative UVB exposure over a more condensed period. These differences may have resulted in faster accumulation of DNA damage, leading to the earlier emergence of detectable lesions in a subset of WT mice.

(3) In addition, biological variability in tumor onset should also be considered. It is important to note that only one WT mouse in our cohort developed a detectable keratoacanthoma by week 25, and this should be interpreted as an isolated early event rather than a generalized shift in tumor latency across the entire group. Inter-individual variability, environmental influences, and subtle experimental differences may also contribute to differences in latency periods.

We hope this explanation helps clarify the basis of the observed differences and addresses the reviewer's concern.

Minor comments

It is unclear whether the skin tumors in Fig.7E are benign or malignant. The authors need to present magnified histological images that clearly show evidence of malignancy (skin tumors growing through the basement membrane and into the mesenchymal layer).

*We thank the reviewer for this comment. To clarify the histopathological nature of the skin tumors shown in Figure 7E, we have included magnified histological images in the newly added **Supplementary Figure 7**, including representative **H&E-stained sections** and **Ki-67 immunohistochemical staining**. These higher-resolution images enable better evaluation of key diagnostic features, including the integrity of the basement membrane and potential dermal invasion. Based on three independent pathological reviews, all tumors were classified as **keratoacanthomas (KAs)** at varying stages of progression. Only one tumor,*

derived from $Wdhd1^{T819A}$ mouse #3, exhibited mild nuclear atypia and focal basal infiltration, suggesting an early cutaneous squamous cell carcinoma (cSCC).

Reviewer #3 (Remarks to the Author):

The authors have addressed all the concerns the reviewer raised. The reviewer would like to support the publication of their manuscript in Nature Communications.

Minor comments:

(1) The authors need to tone down the description of keratoacanthoma (KA): 'KA is currently classified as a low-grade variant of cutaneous squamous cell carcinoma (cSCC) ' (Line 662-663). The website of NIH/NCI (<https://www.cancer.gov/publications/dictionaries/cancer-terms/def/keratoacanthoma>) shows that 'They may look like squamous cell skin cancer, but they are usually benign and rarely spread to other parts of the body' but not cSCC. In the abstract of a recent review (PMID: 39449378), KA is defined as benign tumors (KAs are rapidly growing, benign squamous tumors that are typically well differentiated) but not cSCC.

(2) The authors need to cite PMID: 39449378 on lines 581 and 664.

We thank reviewer for these comments.

We agree with reviewer and have revised the description of KA in the discussion part (Line 676-680) and cited PMID: 39449378 on line 595 and 677.